# ROYAL SOCIETY
# OPEN SCIENCE

statistics/bioinformatics

linguistics, visualization, comparitive statistics

**Author for correspondence:**
Daniel J. Lawson
e-mail: dan.lawson@bristol.ac.uk

# CLARITY: comparing heterogeneous data using dissimilarity

Daniel J. Lawson[1,2], Vinesh Solanki[3], Igor Yanovich[4], Johannes Dellert[5], Damian Ruck[6] and Phillip Endicott[7]

[1]Institute of Statistical Sciences, School of Mathematics, and [2]Integrative Epidemiology Unit, Population Health Sciences, University of Bristol, Bristol, UK
[3]Independent researcher
[4]Department of English and American Studies, Vienna University, Vienna, Austria
[5]Seminar für Sprachwissenschaft; DFG Center 'Words, Bones, Genes, Tools', University of Tübingen, Tübingen, Germany
[6]Department of Anthropology, University of Tennessee, Knoxville, TN, USA
[7]Unité Eco-Anthropologie (EA), Muséum National d'Histoire Naturelle, 17 place du Trocadero, Paris 75016, France

DJL, 0000-0002-5311-6213; DR, 0000-0001-8678-8852

Integrating datasets from different disciplines is hard because the data are often qualitatively different in meaning, scale and reliability. When two datasets describe the same entities, many scientific questions can be phrased around whether the (dis)similarities between entities are conserved across such different data. Our method, CLARITY, quantifies consistency across datasets, identifies where inconsistencies arise and aids in their interpretation. We illustrate this using three diverse comparisons: gene methylation versus expression, evolution of language sounds versus word use, and country-level economic metrics versus cultural beliefs. The non-parametric approach is robust to noise and differences in scaling, and makes only weak assumptions about how the data were generated. It operates by decomposing similarities into two components: a 'structural' component analogous to a clustering, and an underlying 'relationship' between those structures. This allows a 'structural comparison' between two similarity matrices using their predictability from 'structure'. Significance is assessed with the help of re-sampling appropriate for each dataset. The software, CLARITY, is available as an R package from github.com/danjlawson/CLARITY.

## 1. Introduction

The need to compare different sources of information about the same subjects arises in most quantitative sciences. With sufficient effort, it is always possible to construct a model that accounts for data of arbitrary complexity. But without this time-

**Figure 1.** What is CLARITY For? CLARITY compares a *reference* dataset (i) with a *target* dataset (ii–iii) to perform a *structural comparison*. (*a*) *Structure* is defined in terms of the similarity between *subjects* (here letters a–p) that in this example fall into *clusters*. These clusters have a *relationship*, here the distance between clusters. (*b*) The data are quantified as a *similarity matrix* between all subjects. We learn the Structure by minimizing residuals in the reference (*b*-i) and predict the target similarity (*b*-ii,*b*-iii) by relearning the relationship. In (ii) the Structure is the same as the reference but the relationship changes. In (iii), a subject also changes cluster, leading to a structural change. (*c*) This is captured in CLARITY using a residual *persistence chart*. This quantifies how well we predict each subject at a range of representation complexity (here, number of clusters). This allows CLARITY to identify structural change separately to relationship change, for diverse models and data. Box 1 defines the italicized terms and §2.1 discusses this figure.

consuming work, can we visualize the data to determine whether the different sources describe the same qualitative phenomena?

Many datasets are best expressed in terms of similarities or differences between subjects, and are frequently compared by plotting the resulting matrices side by side. Examples include the co-evolution of language and culture [1], as well as genetics and phenotype [2], which are all linked through their geographical constraints and shared history. Further uses include identifying brain function using neural activity patterns [3], understanding disease through comparing the expression of genes with biomarkers [4], toxicology prediction comparing the activation of biological pathways [5] and understanding bacterial function by comparing nucleotide variation to that of amino acids [6].

We describe a new method that is computationally efficient and can be applied whenever similarities or dissimilarities can be defined. We present three diverse examples to demonstrate the potential utility of this method. The first is comparing gene methylation with expression in simulated data containing anomalies; the second compares lexical change with phonetic change data in linguistics, and the third examines the interaction between culture and economics. Beyond providing a new method with extremely wide applicability, this paper aims to focus attention on the problem area of *structural comparisons* in general.

## 1.1. The purpose of CLARITY

Figure 1 is a 'graphical abstract' to illustrate what CLARITY is designed to detect. Rather than learning a model, CLARITY identifies features of one dataset that are anomalous in the second—marginalizing out structures present in both. It does this from information on subjects, i.e. labelled entities on which we have data such as countries, languages, genes, etc.—which are matched across datasets. It looks for

structures—that can be loosely thought of as clusters or shared variance components—that are present in the second dataset, but were not in the first. It reports by identifying subjects with large residuals that 'persist', i.e. cannot be predicted from the first dataset, for a wide range of complexity of representation.

CLARITY works with dissimilarities (and equivalently, similarities), which until they are formally defined (see §4.2), can be thought of as generalizing distances between all pairs of items. Similarities can often be defined even when the data does not form a convenient space, e.g. neural spike trains [7] or genetic relatedness [8]. Similarities are more general than covariances and make a richer representation than a tree—all trees can be represented as a distance, but the converse is not true. They can be defined on regular feature matrices, or on richer spaces, and are robust to the inherent complexity of the data. However, the better chosen a similarity measure is, the better empirical performance can be expected.

CLARITY should have wide application across many disciplines. The paper is written to allow non-specialists to gain insight into the approach and its correct interpretation. Users of the methodology should read the Results and Discussion sections, which include simulated and real examples that should be insightful for specialists and non-specialists of the application area. Further mathematical justification and technical details are available in Methods.

## 1.2. Overview of comparison approaches

How different is the information provided about *the same* subjects in two datasets? For what follows, we are interested in the relationship between the subjects, rather than particular features in the datasets, and we assume that we have enough information to build a meaningful similarity matrix between the subjects.

The gold standard approach involves *generative modelling*, in which the joint model for both datasets is specified. Examples include host–parasite coevolution [9] and comparing linguistic and genetic data [10]. However, each analysis is bespoke, requiring an expert modeller able to specify a joint model for the two datasets.

If the datasets take a matrix form, then *testing* whether two matrices are statistically equivalent is another natural starting point. For this, Mantel's test [11] and related approaches [12], can be used. However, for the sort of scientific investigation that we are considering here, the null hypothesis that the two datasets have 'the same' distribution, or, in the case of shared historical processes, are 'independent' of each other, can often be rejected *a priori*. We are thus interested in richer comparisons, able to highlight specific subjects that behave differently in two datasets.

Data can be directly compared by transforming one to look like the other. When applied to matrices, an important class are *Procrustes transformations* [13], which use rotation, translation and scaling [14] to perform the maximally achievable matching. Procrustes transformations have been used for testing matrix equality under transformation [15] and are often combined with initial rank reduction via *Spectral decomposition* for matrix comparison (e.g. [16]).

If we are not constructing an explicit model of both datasets, nor testing whether they are identical, then the remaining options revolve around constructing summaries that can be compared. Many methods exist to compare *covariance matrices*. Testing [17] is again straightforward. Metrics comparing covariance matrices exist [18], while spectral methods, such as common principal component analysis [19], allow theoretical statements to be made about the results of a comparison [20].

Another important class of summary is *tree-based methods* that represent each dataset as a tree, which can be compared using standard metrics. These include topological distance [21,22] and tree-space [23], and the approach is implemented in popular packages such as 'phangorn' [24] in R. The downside is that handling model uncertainty is difficult, with only some types of tree being stable to small changes in the data [25]. Often the data are not completely hierarchical—for example, tree-based methods can be misleading when the data have a mixture element to them [26]. Conversely, whilst mixtures might be compared using fixed-dimensional mixture-based methods [27,28], this can be misleading when the data have an hierarchical element to them [29].

With CLARITY we are addressing scientific questions that relate to which similarity structures are present in two datasets. There are other scientific questions that might be asked. For example, canonical correlation analysis (CCA) [30,31] and related approaches can be applied on datasets with matched features, as in e.g. ecology [32] and machine learning [33,34]. CCA addresses the question of which features in one dataset are important for understanding another. Because of this focus on features, CCA cannot be used directly in any of the simulations or real datasets that we consider below. Qualitatively, this is because the datasets can match perfectly if the number of features is higher than the number of subjects.

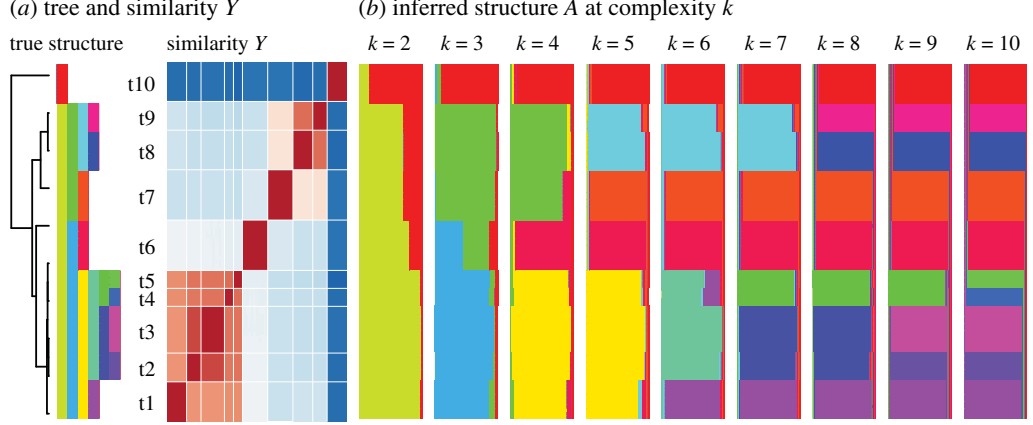

**Figure 2.** What is complexity in CLARITY? CLARITY uses a non-parametric representation that captures a wide class of model, which is interpretable if the truth is interpretable. This is demonstrated with a hierarchical simulated dataset (see §4.6). (a) Subjects are generated as belonging to a cluster under a 'true structure'. Each feature is shared (with noise) with all descendent subjects in the tree, so may represent any branch of the tree, each of which is assigned a colour. CLARITY models the Similarity $Y$ between samples. (b) In such a hierarchical dataset, an inferred mixture Structure at each complexity $k$ relates to a 'soft hierarchy', i.e. a set of mixtures $A_k$ with $k$ components whose latent clusters approximately represent the branches of the tree. Adding components explains different branches, until eventually only noise is explained. (Parameters: $d = 100$, true $k = 10$, $\sigma = 0.005$.)

## 2. Results

### 2.1. High-level view of CLARITY for comparing data from different sources

This section contains a high-level mathematical description of the sort of comparison CLARITY is useful for. Technical mathematics is left for the Methods, §4.2, but at a high level, the key concepts are given in box 1.

CLARITY allows comparison of arbitrary datasets for which the same set of $d$ subjects are observed. It represents the similarity of a *reference* dataset $Y_1$ non-parametrically using increasingly rich representations of *complexity* $k \leq d$. At each $k$ we learn a *structure* $A_k$, which is a $d \times k$ matrix, and a *relationship* $X^{(k)}$ between the structures, which is a $k \times k$ matrix. Learning is by minimizing the squared error of the estimate $\hat{Y}_1$, which is the sum of the squared residuals.

**Example:** *In figure 1b $Y_1$ is a similarity. Figure 1a treats $A_k$ as a clustering learned from the Reference Similarity $Y_1$ (figure 1b). The relationship $X^{(k)}$ is then the similarity between clusters, and the complexity is the number of clusters* k.

We make a *structural comparison* between the reference $Y_1$ and the target $Y_2$, by keeping the *same structure* $A_k$ but fitting a new relationship $X_2^{(k)}$ to make a prediction $\hat{Y}_2^{(k)}$. The procedure is:

1. Construct (dis)*similarity* matrices: both a reference $Y_1$ and a target $Y_2$.
2. Learn structure: Learn $\hat{Y}_1^{(k)}$ in terms of structure $A_k$ and relationship $X_1^{(k)}$ to best predict $Y_1$ for a range of complexities $k$ by minimizing the total error, subject to constraints.
3. Predict conditional on structure: Predict $Y_2$ using $\hat{Y}_2^{(k)}$ which uses $A_k$ at each complexity $k$.
4. Evaluate prediction: examine the residuals $R_2^{(k)} = (Y_2 - \hat{Y}_2^{(k)})$ as a function of $k$. Visually report residuals that are present for many $k$.

**Example, continued:** *In figure 1a $A_k$ are clusterings learned from the reference similarity $Y_1$ with each of $k = 1$, ..., d clusters. We also learn the relationship $X_1^{(k)}$ i.e. similarity between clusters in that dataset. We then predict $Y_2$ from each set of clusters $A_k$, but learn a completely new relationship between clusters $X_2^{(k)}$. With few clusters, we have a bad representation of $Y_1$ while with many clusters we overfit it. We are therefore interested in subjects that are poorly explained for many intermediate clusterings, i.e. those with persistent residuals.*

The CLARITY model uses a range of *complexities* $k$ to represent data. The full set of structures therefore quantifies a rich range of models. For example, it can be constrained to a hierarchical clustering (i.e. a tree), and in §2.4 we show that if the data are tree-like, the structure can be interpreted in terms of a 'soft tree' (figure 2), which can capture deviations from a strict tree model.

**Example, continued:** *In figure 1c, persistencies at different complexities $k$ are plotted. Observe that at high $k$, residuals and consequently persistences largely perish. In (ii), we have the same clustering (i.e. structure) but a different relationship, and in c(ii) the large decrease in persistences happens at $k = 4$, i.e. as soon as complexity $k$ reaches the true number of clusters in the data. In contrast, in c(iii), the data are generated under a clustering*

**Box 1:** Important concepts in CLARITY.

**Similarity $Y$:** A $d \times d$ matrix comparing all subjects, for which 'closer' subjects have large pairwise values. A typical example is a covariance. CLARITY operates entirely equivalently with dissimilarities for which 'close' implies small values; a typical example is the Euclidean distance.

**Structure $A_k$:** A $d \times k$ matrix, providing a representation of each subject in $k$ dimensions. The choice of *structure* is the most important choice in CLARITY. It is learned subject to constraints on $A$, conditional on the relationship. Three important Structures are (i) a clustering, with entries of $A$ being 0 except when subject $i$ is in cluster $k$ in which case $A_{ik} = 1$; (ii) a Mixture: rows of $A_{ij}$ sum to 1 and $A_{ij}$ are non-negative; and (iii) an unconstrained subspace, in which case $A$ is the top $k$ eigenvectors, as seen in spectral methods (singular value decomposition, SVD and principal components analysis, PCA).

**Relationship $X^{(k)}$:** The *relationship* between structures depends on the structure. We again learn it by minimizing the error conditional on the structure. In this paper we do not consider constraints. For a clustering, the relationship is the similarity between clusters, or the similarity between latent clusters in a mixture model. It describes the 'branch lengths' of a tree. For PCA, the relationship is a rotation, translation and scaling of the matrix of singular values.

**Structural comparison:** The *structure* learned from the reference $Y_1$ at each complexity is used to predict the target $Y_2$. $Y_2$ may be numerically quite different if the relationships are different. However, as long as the datasets can be predicted in this sense then we say that the matrices are defined to be *structurally similar*; this will happen if the same clusters, mixtures or eigenvalues are important in both datasets and describe the same subjects.

**Residual persistence charts:** We use graphical summaries to present useful scientific insights, focusing on structures that persist over a range of model complexity. These are inspired by the concept of persistent homology from Topological Data Analysis [35]. When the complexity $k$ is sufficiently high, every (full rank) dataset can predict every other, so the focus is on which structures in $Y_2$ are explained late in the sequence defined by $Y_1$, i.e. persist. This is captured by the residual persistence $P_{ik}$, a matrix whose entries for data subject $i$ at a complexity $k$, are the sum (over $j$) of the squared residuals.

*that differs by one subject from the structure (i.e. the clustering) from the first dataset. In this condition, persistences remain high until much higher complexity, especially so for the anomalous subject i.*

**Key decisions when using CLARITY:** There are just two substantive choices to make when running CLARITY. First, how to construct the similarity matrices $Y_1$ and $Y_2$, which may be determined by the data. Second, the choice of structure $A_k$, for which the two main approaches we provide are SVD, which is computationally convenient, or a mixture model, which is more interpretable. We have not found a case for using clusters in practice, as they are not robust. There is one final technical choice, which is the statistical procedure for assessing significance, which we address in §4.4. Good practice for all choices is discussed in §3.1.

## 2.2. Learning structure in CLARITY

Two similarity matrices are structurally similar if one can be predicted from the other, using the partial representation we have defined as *structure*. CLARITY is comparing similarity matrices $Y$ which requires a *quadratic* model (in $A$) rather than the more familiar linear model

$$\hat{Y} = AXA^T,$$

where $A$ and $X$ are intended to be 'simpler' (concretely, $k$-dimensional) approximations to $Y$.

**Example, continued:** *When A is a clustering, then rows of A are subjects and columns are clusters. Row i of A is a vector with value 1 if subject i is in cluster j. We then predict the similarity $Y_{ij}$ with an estimate $\hat{Y}_{ij}$ between subject i and j by finding which cluster $c_i$ and $c_j$ each is in, and replacing it with the similarity between those clusters, $X(c_i, c_j)$.*

We always seek to minimize a loss $L(A, X)$ which is the (squared) error. Setting $R(i, j) = Y(i, j) - \hat{Y}(i, j)$, the loss is

$$L(A, X) = \sum_{i=1}^{d} \sum_{j=1}^{d} R(i, j)^2.$$

If this minimization is unconstrained, then (see Methods §4.2) $A$ is the matrix of the (first $k$) eigenvectors and $X$ the diagonal matrix of (first $k$) singular values of $Y$. $Y_2$ is structurally similar to $Y_1$ if it can be predicted using this learned $A$ and a new $X_2$. Technically, if $Y_2$ is poorly predicted at complexity $k$ then it is not close to the subspace spanned by the first $k$ principal components of variation in $Y_1$.

When $A$ is constrained to be a mixture (that is, its elements are non-negative and sum to one), $X$ describes the similarity between 'latent clusters', and the rows of $A$ describe mixtures between these clusters. Similarity matrices are hence 'structurally similar' if they can be described by the same mixture. This mixture model is interpretable, as we demonstrate in the simulation study. Specifically, a 'structural difference' at complexity $k$ means that the latent clusters of $Y_2$ are not in the $k$ most important latent clusters of $Y_1$. Further, the subjects that are poorly predicted, i.e. whose cluster membership is not captured, can be read off from the residual matrix.

Persistences and residuals decrease with model complexity and are affected by correlations between similarities. Despite the complexities of working with a similarity matrix, in theorem 4.2 (Methods §4.8) we prove that the model is stable in the presence of noise, so that if two datasets were resampled then their residuals based on our notion of structure are not expected to change by a large amount. The theoretical and simulation results together demonstrate that the CLARITY paradigm is performing a meaningful comparison.

## 2.3. Structure of the examples

We now present use cases. First we consider a simulation study for a mixture of trees model in §2.4, to demonstrate how CLARITY can be used to identify differences in large-scale structure. In §2.5, we consider simulated data based on methylation and gene expression data to give an example in which CLARITY can be used for anomaly detection. Section 2.6 is a real-data example from linguistics, which is smaller in scale and therefore more subtle in interpretation. Finally, §2.7 examines the relationship between culture and economics at the country scale. The take-home messages from these examples are summarized in §3.1.

## 2.4. Comparing simulated hierarchical mixtures

Hierarchical data are common and naturally interpretable using CLARITY. In this section we simulate subjects related by a tree and insert an interpretable structural difference between two datasets. The relationship between structures includes features such as the branch lengths of the tree. The structure itself is defined by the membership of subjects in the clusters. Both are detectable with CLARITY but changing structure creates a much larger effect in the data.

### 2.4.1. Simulation model

The model creates data that are generated with $N = 100$ subjects observed at $L = 2000$ features, grouped into $k = 10$ clusters related via a tree. These data are used as a reference for a CLARITY model. In *Scenario A*, we construct a target by regenerating the tree with the same topology but altering the branch lengths, and resimulating data. In *Scenario B*, we construct a target with the branch lengths changed as in Scenario A, but additionally change the structure $A$. See §4.6 for details.

### 2.4.2. Hierarchical mixture inference

Figure 1 shows that CLARITY is insensitive to changes in the heatmaps themselves, but remains sensitive to changes in structure. Figure 2 shows how the CLARITY mixture model infers detailed structure of $A_k$ capturing the clusters present in the data when the tree is 'cut' at different heights. Figure 3 illustrates how this is achieved.

When we use the $Y_1$ structure (figure 3$a$) for prediction of the second similarity matrix $Y_2$ (figure 3$c,e$), several situations may occur. In Scenario A-1, the trees share the same split ordering, and any differences will be completely absorbed by differences between $X_1$ and $X_2$. The residuals and persistences of $Y_2$ will be distributed as for $Y_1$ (see §4.4 for how this is estimated). In scenario A-2, the trees share the same topology but the split order differs. The structures in $Y_2$ may not appear in exactly the same order as in $Y_1$ but still all appear in the top $k < k_{\max} = 10$ structures representing the tree. The residuals and persistences may be larger at lower complexity, as happens in figures 3$c,d$ and 1$c$, but the entire

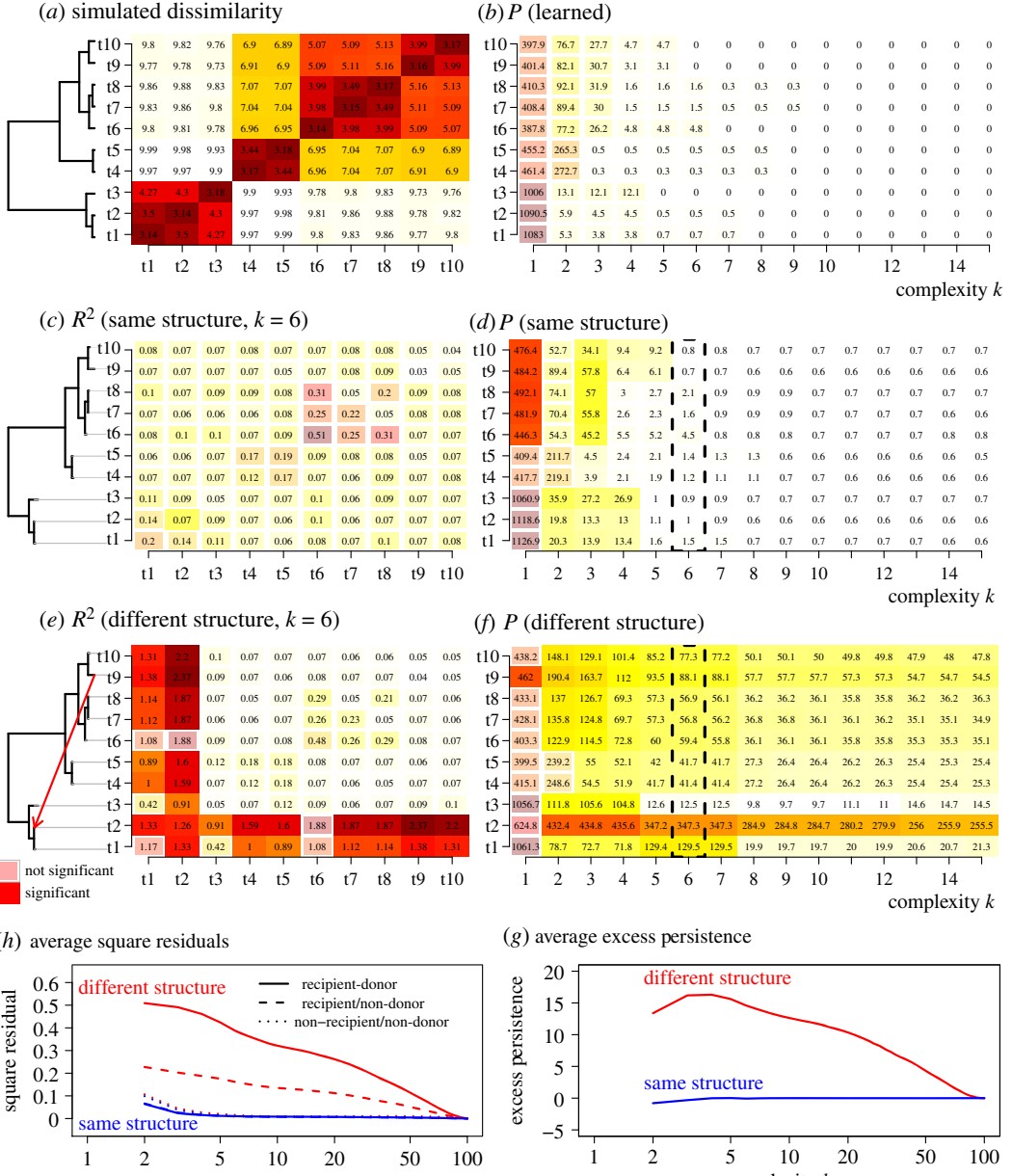

**Figure 3.** Interpreting residuals and persistences using simulated data, for $d = 100$ subjects from $k = 10$ clusters. Dissimilarities shown are averaged within clusters, whilst residuals and persistences are summed to create population values. (*a*) Learned tree and dissimilarity matrix. (*b*) Residual persistence chart for the learned data. (*c*) Squared residuals and (*d*) residual persistence, for new simulated data with the same structure as in (*a*). (*e*) Squared residuals and (*f*) residual persistence, for simulated data with a different structure to (*a*), for which some subjects in cluster t2 are a mixture with t9. For (*c*–*f*) lack of significance at $p = 0.01$ is illustrated by drawing a smaller rectangle. (*g,h*) Replicated results averaged over 200 simulations. (*g*) 'excess persistence' which is the residual persistence of samples in the recipient cluster—i.e. t2 in the tree in (*e*)—with the mean residual persistence of the other samples subtracted. (*h*) Summed squared residuals for different parts of the residual matrix. Shown is the 'recipient' compared with the 'donor'—t9 in (*e*)—as well as the recipient compared with all non-donor samples, and the average residuals for all pairs of samples that were neither recipient nor donor. Simulation settings: $\sigma = 0.05$, $\beta = 0.5$, for which in 'different structure', half of the recipient cluster is affected by the mixture.

difference can be explained at some complexity threshold. Things are different in Scenario B when $Y_2$ has a different topology to $Y_1$—perhaps containing mixtures as in figure 3*e,f* or new clusters, such as figure 1*a*. Only then will important structure be absent until much higher $k$ and this will result in high *and persistent* residuals for the affected data (figures 3*c* and 1(iii)).

The persistence $P$ in figure 3*f* identifies the clusters that are affected by the structural change: cluster group t2 has significantly inflated $P$. Examining the residuals themselves at a specific $k$, figure 3*e*

identifies the two clusters affected, which have highest off-diagonal shared residuals. In addition, they show that the 'recipient' cluster t2 has consistently high pairwise residuals. The 'donor' cluster t9 does not have exceptional residuals overall, but does have the highest pairwise residual with the 'recipient' t2. Of note is that low-dimensional representations ($k = 1$, 2) are not helpful because there is high intrinsic variability (i.e. these persistences are large but not significant). We must have a 'good enough model' of $Y_1$ before it is useful to understand $Y_2$.

This interpretation is robustly replicated in simulations, as is shown in figure 3$g,h$ for 200 different mixture-of-tree simulations. Specifically,

— Persistence is high in 'recipient' clusters of $Y_2$ containing a mixture of two different signals of the structure found in $Y_1$.
— Squared residuals of the recipient cluster are high with all clusters that are topologically close to affected subjects. This happens both under the original structure and the 'new' structure in which the 'recipient' and 'donor' clusters are close.
— Persistence for 'donor' clusters is not exceptional, but they are identifiable from their very high residuals with the recipient cluster.

In this way, the residuals for tree-like data can be interpreted topologically by first identifying clusters of subjects that experience a high residual persistence. The source of the mixture can be identified from which clusters (that are dissimilar in the reference) have increased residuals with these subjects.

Whilst the Mixture model allows interpretation of how structural differences can be detected, both the Mixture model and SVD model make comparable predictions. Electronic supplementary material, figure S1 shows that the *same* structural similarity information is learned from the SVD model as in the Mixture model. The models predict $Y_1$ with near identical performance. Further, they both agree that the presence of different structure leads to poorer prediction of $Y_2$ from $Y_1$ for a wide range of $k$.

In terms of computational complexity, the SVD method is dominated by the SVD ($O(d^3)$). For reference, it takes 6.75 min to run our SVD model for $d = 5000$ on a personal laptop, most of which is computing the SVD. The mixture model is dominated by a $d \times d$ matrix inversion ($O(d^3)$ or better) but is in practice slower as the convergence time of the iterative algorithm scales with $d$.

## 2.5. Comparing simulated gene methylation to gene expression

This example focuses on identifying *anomalies*: subjects (here loci) that behave differently in one dataset compared with another. The CLARITY framing using a similarity matrix means that the datasets need not have features in common.

It is very common in epigenetics to wish to compare different measurements on genes. These are often measured on different scales—methylation is a proportion whilst expression is a positive number—and writing a formal model is hard. Further, the raw data may not be available to each researcher as they are identifiable and sensitive. It is therefore very helpful if summaries, such as those based on similarities, can be compared.

To emphasize the utility of CLARITY we focus on an epigenetic simulation model from the literature [36], in which Methylation and Expression data are generated from *independent* case-control experiments that describe the same set of genetic loci, i.e. positions in the genome, that correspond to known genes. These loci are our subjects. We do not claim that this is a realistic model, only that it has received attention.

Methylation is a chemical process that reduces the accessibility of DNA for transcription, and is therefore negatively associated with gene expression. It can be measured using high-throughput arrays (e.g. [37,38]) that target known methylation loci. Similarly, gene expression can be measured using high-throughput arrays by capturing the transcribed RNA [39]. However, both approaches are subject to a variety of noise, including inter-cell and inter-subject variability, high stochasticity in measurement from the amplification process, and so on. This variability adds to signal from causative factors of interest. The simulation therefore assumes only a −5% average correlation.

The simulation is for a case-control study in which some people were controls, and others had a tumour. Methylation is simulated in different classes ('hyper' and 'hypo' methylated loci) which interact with case-control status. Expression is simulated conditional on methylation, with large and random noise. Finally, anomalous loci are chosen in which the relationship between methylation and expression is reversed. See Methods §4.5 for full details.

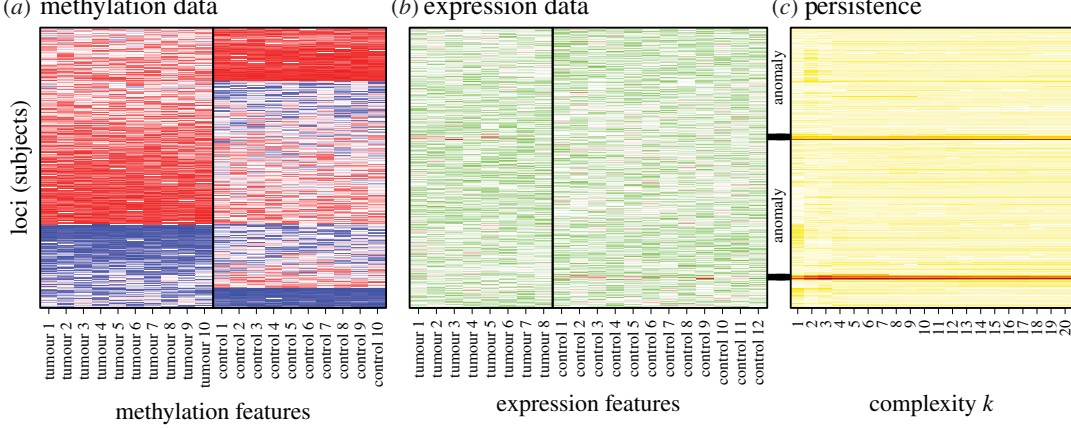

**Figure 4.** A simulated Methylation/Expression heatmap comparison from [36] with added anomalies (see Methods §4.5). (*a*) A reference dataset is used to learn structure; here, simulated Methylation patterns across the genome. (*b*) The structure is used to predict the similarities in the target dataset; here, gene expression data at the same loci, with inserted anomalies. The data need not describe the same features (here, samples with Tumour/Control status). (*c*) *Persistent residuals* over a range of *l* indicate which subjects (here, loci) are structurally different between the target data and the reference data, accurately identifying the anomalies.

The important feature of this set-up is that the set of people considered to quantify methylation and expression are independent. The only thing they share in common is the set of loci on which data are reported. Figure 4 illustrates this dataset and highlights the ability of CLARITY to extract the anomalous loci, despite the considerable structure in this dataset.

Figure 4 shows the two input datasets $Y_1$ and $Y_2$, which contain some clear visual structures, and specifically loci that differ between Tumour and Control. It then shows the *persistent residuals* as estimated by CLARITY. A subset of these are 'anomalies', one cluster of which (top) has lower expression than expected in Tumour samples, the other has lower expression in Controls. Both are clearly highlighted via the Persistence chart (figure 4*c*) over a range of *k*, even though the changes are individually small.

## 2.6. Two types of language change

In this example, we also search for *anomalies*, but in contrast to the previous case, our finding here is both surprising and scientifically significant. It therefore merits explicit hypothesis testing, as well as a number of further checks making sure that the effect we find is not spurious. We compare two different, but non-independent types of language change: phonetic, or sound, change; and lexical, or word-replacement, change. There is no question that sound (i.e. phonetic) change and word-replacement (i.e. lexical) change are correlated because they result from a shared historical process: both types of change are inherent to the transmission of language from one generation of speakers to the next. There are two general sources for both phonetic and lexical change. First, each language changes on its own as time proceeds, even in complete isolation from external influences [40]. Second, languages can influence each other when there are multilingual people, with this process being called *language contact* [41].

In the linguistic literature, it is often argued qualitatively that phonetic and lexical change can be unevenly favoured by different social situations of language transmission and contact (e.g. [42]). Here, we use CLARITY to provide a quantitative test, the first such experiment known to us. Figure 7*a*,*b*, demonstrates that the two change types induce closely aligned similarities between language. While this is not surprising, given that the same factors influence both types of linguistic change, we show that, despite this high degree of correlation, there are still detectable 'structural' differences between lexical and phonetic changes. Two components of our comparison are crucial: (i) we use a large dataset that enables us to apply CLARITY significance testing and thus derive a statistically credible result; (ii) allowing the CLARITY 'relationship' to differ for phonetic and lexical, we effectively permit the rates of both types of change to differ considerably without the matrices becoming structurally dissimilar; in other words, the null hypothesis of CLARITY is broad enough that its rejection would be informative.

The true history of both phonetic and lexical change is unknown, but can be conceptualized as a graph that consists of a 'vertical' inheritance via a tree that captures independent change, and 'horizontal' edges that capture change through language contact. Such a graph induces a similarity matrix between

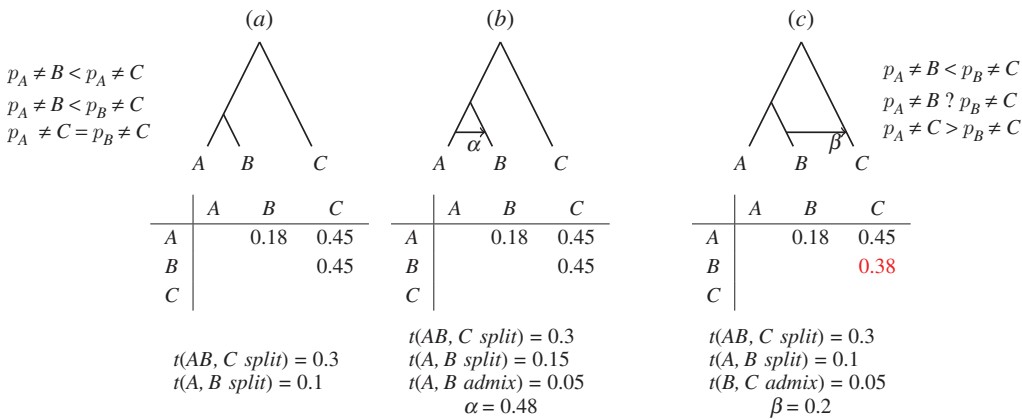

**Figure 5.** Some history-of-change graphs are not distinguishable from dissimilarity matrices, and others are. Let each of $A$, $B$ and $C$ be a feature descending from the root node, with a constant rate of change. Then the probability of a mismatch between $X$ and $Y$, $p_{X \neq Y}$, is a weighted sum of terms $1 - e^{-t}$, where $t$ is the length of a path between $X$ and $Y$ in the graph, and the weights are given by the probabilities a given path was taken, determined by admixture proportions $\alpha$ and $\beta$ in ($b$,$c$). Graphs ($a$,$b$) differ, but with specific values for the times of splits and $\alpha$ can lead to exactly the same probabilities of mismatch, and thus the same dissimilarity matrices. Graph ($c$), in contrast, leads to a different dissimilarity matrix over any choice of split times and $\beta$ as long as $\beta$ is not 0.

languages. Many graphs may induce the same matrix, making direct inference of the history graph impossible in the general case, figure 5$a$,$b$. But the similarity matrix does allow us to distinguish between different *classes* of graphs (cf. figure 5$a$–$c$). With CLARITY, we can infer changes to the structure of the underlying graph without explicitly learning that graph. With two matrices representing similarity due to phonetic versus lexical change, we can use CLARITY to find out whether phonetic and lexical change go hand in hand. Our null hypothesis is that lexical and phonetic change are aligned, because they ultimately stem from the same interactions between and within speech communities. It is the rejection of this null hypothesis that would be scientifically interesting. This is an appropriate set-up for applying CLARITY, which looks for evidence of differences between two (dis)similarity matrices.

In the limit of an infinite number of linguistic features, there exists a 'true' matrix induced by the history-of-change graph. But in practice we have to work with matrices estimated from a finite amount of data. To achieve a reasonable estimate, we need many individual features, which in practice requires automatic methods for inferring both phonetic and lexical similarity. Automatic methods incur an inherent error at the level of individual features, but processing many features results in reasonably stable estimates of similarity matrices. Specifically, in cross-validation, we show that however we divide our features into two halves, the halves can predict each other with great success, i.e. carry very similar information.

Furthermore, an important technical detail is that our automatic lexical-similarity recognition operates on word-to-word phonetic similarity scores. This dependence amplifies the correlation between the lexical and phonetic similarities, thereby working in favour of the null hypothesis. This way, we can be confident that a rejection of the null hypothesis would correspond to a true real-world difference in 'structure'. Finally, automatic cognate detection cannot distinguish between cognates inherited from the last common ancestor and words that have been borrowed between sister branches after divergence. For recovering the true tree part of the language-family history, this is a drawback; for us, this is a benefit, as it captures language contact relations in the lexical-change data, just as it affects the phonetic-change data.

Our data come from one of the largest existing historical-linguistic datasets, NorthEuraLex v. 0.9 [43], which stores phonetic transcriptions of words expressing 1016 different meanings in over a 100 languages. We focus on the 36 Indo-European languages in NorthEuraLex, for which we computed measures of both phonetic and lexical dissimilarity using a state-of-the-art method [44], as discussed in more detail in §4.7.

Tables in figure 6 illustrate how the phonetic and lexical similarities work in practice, using 10 meanings and words for their expression in English, Danish and German. Historical linguists have established that English and German have a more recent common ancestor than either has with Danish (i.e. in the vertical 'backbone tree'). After their respective divergences, the three languages experienced complex language-contact patterns, which can be conceptualized as horizontal edges in their historical graphs.

We discuss three rows in figure 6 to illustrate the individual-feature diversity behind the general similarities. The meaning BEARD in figure 6$a$ is straightforward. The English and German words

(a)

| meaning | EN | DA | DE | ph:EN-DA | ph:EN-DE | ph:DA-DE | le:EN-DA | le:EN-DE | le:DA-DE |
|---|---|---|---|---|---|---|---|---|---|
| BEARD | beard | skæg | Bartt | 0.27 | 0.523 | 0.27 | no | yes | no |
| HEAD | head | hoved | Kopf | 0.54 | 0.068 | 0.34 | yes | no | no |
| GREEN | green | grøn | grün | 0.43 | 0.553 | 0.687 | yes | yes | yes |
| DOG | dog | hund | Hund | 0.23 | 0.119 | 0.697 | no | no | yes |
| FIRE | fire | ild | Feuer | 0.17 | 0.539 | 0.166 | no | yes | no |
| COME | come | komme | kommen | 0.72 | 0.585 | 0.712 | yes | yes | yes |
| THROW | throw | kaste | werfen | 0.23 | 0.231 | 0.182 | no | no | no |
| SLEEP | sleep | sove | schlafen | 0.27 | 0.338 | 0.288 | no | yes | no |
| DRINK | drink | drikke | trinken | 0.47 | 0.566 | 0.491 | yes | yes | yes |
| TAKE | take | tage | nehmen | 0.40 | 0.302 | 0.183 | yes | no | no |

(b) phonetic from 10 meanings

| phon | EN | DA | DE |
|---|---|---|---|
| ENG | 1.000 | 0.377 | 0.382 |
| DAN | 0.377 | 1.000 | 0.402 |
| DEU | 0.382 | 0.402 | 1.000 |

(c) lexical from 10 meanings

| lex | EN | DA | DE |
|---|---|---|---|
| EN | 1.000 | 0.500 | 0.600 |
| DA | 0.500 | 1.000 | 0.400 |
| DE | 0.600 | 0.400 | 1.000 |

(d) phonetic from full data

| phon | EN | DA | DE |
|---|---|---|---|
| EN | 1.000 | 0.312 | 0.318 |
| DA | 0.312 | 1.000 | 0.458 |
| DE | 0.318 | 0.458 | 1.000 |

(e) lexical from full data

| lex | EN | DA | DE |
|---|---|---|---|
| EN | 1.000 | 0.399 | 0.415 |
| DA | 0.399 | 1.000 | 0.593 |
| DE | 0.415 | 0.593 | 1.000 |

**Figure 6.** (a) Phonetic and lexical word-to-word similarities for 10 meanings in English (EN), Danish (DA) and German (DE). Columns ph:A-B list inferred phonetic similarity between languages $A$ and $B$ accounting for regular sound correspondences. Columns le:A-B list word-to-word lexical similarity, namely cognacy (i.e. descending from the same ancestral word), inferred via clustering based on phonetic similarity scores as in ph:A-B. See §4.7 for details of the measures. (b,c) The phonetic and lexical similarity matrices induced by the data in (a). (d,e) Same, induced by the full data.

descend from the same ancestral word, experiencing no lexical-replacement events. In linguistic parlance, these words are cognates. This common ancestry means that their phonetic shape descends from one and the same shape of the ancestral word, and thus causes them to be phonetically similar in the real world. Our phonetic-similarity algorithm correctly recovers that information (column ph:EN-DE), and based on this high level of similarity, our lexical cognate-detection algorithm also declares them lexically similar ('yes', or 1, in le:EN-DE). The Danish word for BEARD is historically unrelated, and is indeed not similar to either German or English word phonetically. For the meaning HEAD, it is the English and Danish words that are true cognates, and are inferred to be both phonetically and lexically similar by the algorithms. But the German word for HEAD is accidentally much closer phonetically to the Danish word than to the English one. The more the phonetic systems of two languages are alike, the more frequent and pronounced such accidental similarities of unrelated words will be on average. Finally, the meaning GREEN illustrates a different property of phonetic similarity. For GREEN, all three languages use cognate words, and are correctly inferred to do so by the lexical algorithm. But their phonetic similarities to each other differ. The English vowel in 'green' is closer to that in German 'grün' than Danish 'grøn'. For cognate words, phonetic similarity depends on how sound changes operated on what was initially one and the same word.

Schematically, our automatically inferred similarity measures Phon and Lex can be characterized as follows. Lex = TrueLex + $\epsilon_{lex}$, where TrueLex is the true lexical similarity, and $\epsilon_{lex}$ is the error from the automatic cognacy-detection algorithm. Phon = TrueLex∗unrelated.phon.sim + (1 − TrueLex)∗cognate. phon.sim + $\epsilon_{phon}$, where we condition on whether the relevant words are related (with TrueLex impressionistically viewed as probability) or not; unrelated.phon.sim and cognate.phon.sim are two types of phonetic word-to-word similarities discussed in the previous paragraph; and $\epsilon_{phon}$ the error of the phonetic algorithm. This informal representation shows how our empirical measures Phon and Lex are even more dependent than the real-world quantities TrueLex, unrelated.phon.sim and cognate.phon.sim. In turn, the latter three are also dependent because they are the result of language-change processes in the same speaker communities.

Figure 7a,b shows our similarity matrices for Phonetic and Lexical, which look very much alike. Figure 7c,d shows the CLARITY analysis, which uncovers structural differences between the matrices in (a,b). The differences between Phonetic and Lexical are declared *statistically significant* by the CLARITY hypothesis-testing procedure based on resampling. In the figure, significant persistences are marked by a brighter colour and a larger rectangle. They only concern one direction of prediction,

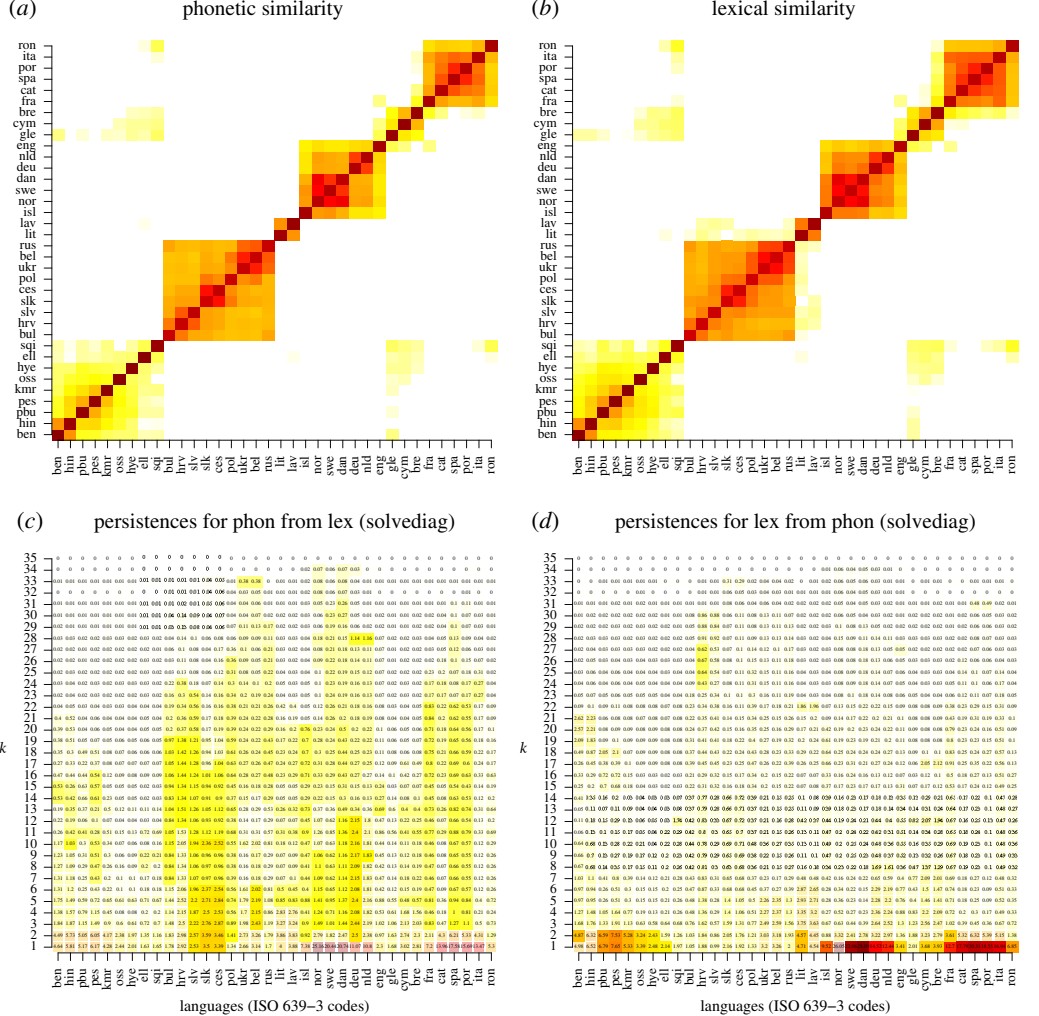

**Figure 7.** (*a,b*) Phonetic and Lexical similarity matrices. The three large orange-red clusters correspond, left to right, to the Slavic, Germanic and Romance language subfamilies. (*c,d*) Persistence diagrams. Statistical significance for cells is visually highlighted by colour saturation and by the size of the background rectangle. Compare the columns for `oss` (Ossetian) and `slv` (Slovenian) in (*c*). The column for `oss` has no significant cells. The column for `slv` has significant cells up to $k = 23$ and non-significant ones higher than that.

namely Phon from Lex, and only some groups of languages; the most affected ones are the Slavic and Scandinavian subfamilies. The differences are also *scientifically significant*, which we determined by analysing the residuals at individual $k$, shown in electronic supplementary material, figure S2: outstanding residuals remained of the order of 0.1–0.2 s.d. (computed from all similarities in the matrix) in individual cells even at the highest $k$s. We believe this is a moderate, yet considerable difference. Our additional checks also included establishing that CLARITY decomposition captures signal rather than noise up to the maximal $k$ (electronic supplementary material, figure S3); checking whether self-similarity affects inference (electronic supplementary material, figure S4) examining significances at a stricter $p = 0.01$ (electronic supplementary material, figure S5) and examining the persistence curves for the resampled matrices to make sure there were no anomalies (electronic supplementary material, figures S6 and S7). We conclude that the effect CLARITY finds, captured in the visual summary in figure 7*c,d*, is a real one.

This result obtained by CLARITY is striking because it is based on very subtle distinctions in the observed data. To the bare eye, the Phonetic and Lexical distance matrices figure 7*a,b* are quite similar, because the two sets of features are not independent of each other. Despite high correlation, CLARITY allowed us to discover a clear difference between the two processes of language change.

How should we interpret this finding? In terms of the informal representation above, our results about Lex and Phon indicates that the real-world quantities TrueLex on the one hand, and unrelated.phon.sim and cognate.phon.sim on the other, are affected by subtly different historical

processes. Given that Phon depends on a superset of real-world quantities that Lex depends on, it is not surprising that we only find the effect in the direction of predicting Phon from Lex. The difference between the two is considerable, but is only discovered with statistical significance for groups of languages closely related to each other, such as the Slavic and Scandinavian languages. This might be a real-world phenomenon: perhaps in the long run, the phonetic and lexical change processes average out and start looking very similar). It could also be an artefact of our algorithms for estimating similarity: as all computational-historical-linguistic algorithms, they inherently tend to be more accurate for more closely related languages, so it might be that we see significant mismatches between Phonetic and Lexical only where our estimates can be sharp enough. We leave solving this question to future research.

We conclude this case study with the following strong thesis: qualitative linguistic research into small sets of linguistic features should refrain from generalizing its results to the overall workings of different types of change. The statistical mismatch between Phonetic and Lexical that we found was subtle and required a large dataset to be discovered.

## 2.7. Predicting culture from economics

In this example, we use the 'structure' from economic properties of countries to predict cultural values. The purpose of this case study is to demonstrate how exploratory analysis with CLARITY can be performed on the level of individual anomalous residuals. Overall, country-level economics and culture similarity matrices are significantly different, as is to be expected. But examining the CLARITY residual structure, we identify particularly interesting anomalies.

For Economics, the World Bank [45] provides a range of features primarily relating to wealth, inequality, trade and the like for over 200 countries. The World and European Values Surveys (WEVS) [46,47] provides features on Culture—that is, people's attitudes and beliefs regarding topics like religion, prosociality, openness to out-groups, justifiability of homosexuality, political engagement and trust in national institutions. There are 104 countries shared between these data that represent our subjects for this case study. The 'Cultural' dissimilarity matrix, figure 8a is constructed from a dimensionality-reduced dataset of nine cultural factors from WEVS from ca 2000 CE, as described by Ruck et al. [48]. For 'Economics', we retained the 284 indicators of the World Bank dataset with less than 40% missingness, standardized to unit variance, capped extreme values at 10 s.d., mean imputed, and computed the 'Economic' pairwise distance matrix. The raw dissimilarity matrices are shown in electronic supplementary material, figure S8.

Cultural values are known to predict economic outcomes such as GDP *per capita* [48,49], economic inequality [50] and the balance of agriculture-industrial-service sectors within the economy [51]. Conversely, a country's economic performance predicts cultural factors such as religiosity [52] and book writing [53]. CLARITY does not presuppose a causal model and therefore the choice of reference and target should not be interpreted as a causal claim without additional information.

Because we have only nine features for Culture, it cannot be used to identify persistent residuals in Economics with CLARITY as the maximum complexity is $k_{max} = \min(k, d) = 9$. Instead we ask whether Culture (figure 8a) can be predicted from Economic data, in which $k_{max} = d = 104$. The persistence chart (figure 8b) makes it clear that Culture is incompletely predicted from Economics, as almost all Persistences are significant. Conversely, not all residuals are significant. The residual matrix at a specific $k$ (figure 8c) identifies the most important mismatches between Culture and Economics, which may be worthy of further study using other methods and/or data. There are two main classes: anomalies and clusters.

One anomaly is Andorra, a small European country in the Pyrenees between Spain and France, which belongs culturally, to the cluster of Scandinavian and economically strong Western European countries, figure 8a. But predicting Culture from Economics in figure 8c, we see highly negative (blue) residuals between Andorra and its cultural cluster. (Remember that absence of significance simply means we do not have *enough* evidence to reject the null hypothesis based on the available data.) Examining the raw data in electronic supplementary material, figure S8, shows Andorra to be an Economic outlier, clustering with other territories with complex sovereignty that may influence data gathering: Taiwan, the Turkish region of Cyprus, and Northern Ireland.

Now consider Vietnam and Uzbekistan. Their columns of residuals in figure 8c are all green, meaning they are farther away on Culture than expected from Economics from all (!) other countries. These two countries are the only ones with all-green residual columns in figure 8c. Examining the raw data in figure 8a and electronic supplementary material, figure S8, as well as the Economic PCs in electronic supplementary material, figure S9, we see that they are not particularly close in either Culture or Economics to any other countries, but that there is little relation between which countries are closer in

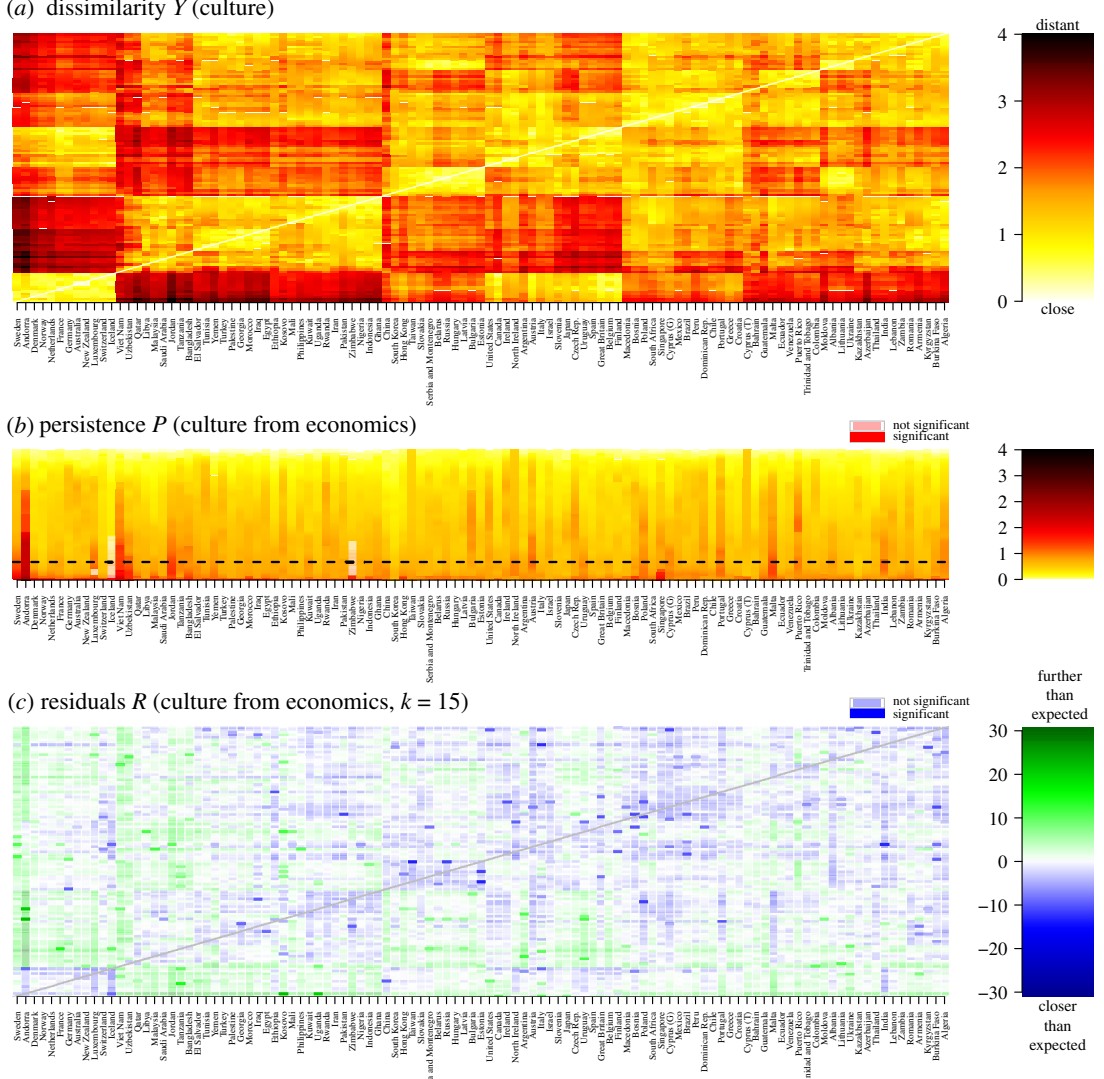

**Figure 8.** Predicting economic properties from cultural properties. (*a*) The dissimilarity $Y_2$ for Culture. The order of countries is determined by clustering based on the dissimilarity, so countries that are close on the *x*-axis are, other things being equal, close to each other in the Culture dataset. (*b*) Persistence of Culture predicted from Economics. The dashed line indicates the complexity *k* used in (*c*). (*c*) Residuals for Culture predicted from Economics at $k = 15$.

either. Future research may thus want to first confirm that the cultural unusualness is not spurious (based e.g. on coding errors and the like in the datasets as processed), and second, to study what could be driving these anomalous profiles.

There are countries that share residual structure. Countries in Latin America, such as Argentina, Uruguay and Puerto Rico, have large European descended populations [54,55], so are culturally similar to Europe because cultural values percolate along linguistic and religious pathways [56–58]. However, Latin American countries have a relatively smaller GDP and have high economic inequality; they are therefore more culturally similar to Europe than Economics predict.

The Cultural data contain associations that are perhaps surprising; for example, Japan is culturally similar to the Czech Republic. Correcting for Economics, figure 8*c*, reveals that this cannot be fully explained away by Economic similarity (the relevant cell is light blue, though not significant). In contrast, Poland and the Czech Republic are neighbouring countries that cluster very differently in Culture, and have large persistences. In this case, Economics may be playing an even smaller role than in the comparison Japan–Czech Republic: the residual Poland–Czech Republic is significant and shows they are much farther from each other on Culture than we could expect from Economics.

To conclude, we have described how CLARITY analysis identifies interesting anomalies in one dataset that stand out when making predictions based on the other dataset. In this case, the prediction is poor, so considerable follow-up analysis would be needed to draw strong conclusions.

The assessment of statistical significance was of secondary importance: we concentrated on systematic patterns in the CLARITY residuals that are likely to be of scientific significance, and the presence of statistically significant individual cells only confirmed the insights. This strategy is particularly useful for datasets based on a relatively small number of features, which will often be the case in social sciences: the statistical power of such data would be in the general case limited.

## 2.8. Summary of the examples

We summarize the lessons from the four case studies above. In §2.4, we show how CLARITY behaves on data from a simulation where we purposefully manipulated the 'structure' of the compared datasets. This case study builds intuitions about what CLARITY output would look like in different real-world scenarios. In another simulation study in §2.5, we demonstrate that CLARITY correctly identifies anomalous links between two datasets generated by an existing epigenetic model that did not conform to our definition of Structure. In the linguistic study on real-world data in §2.6, we report the first-of-its-kind quantitative finding that phonetic and lexical language change operates in subtly different ways, though both occur in the same human communities and are subject to similar constraining factors. This example is the only one in this paper where explicit hypothesis testing is of primary importance. Finally, in the example on culture and economics in §2.7, we illustrated how CLARITY residuals can be explored to detect interesting anomalies that can then be selected for further in-depth study.

Overall, we intend this set of examples to show how CLARITY can be a versatile tool for both exploratory data analysis and for explicit hypothesis testing, allowing us to make subtle and fine-grained comparisons between paired datasets stemming from the same subjects, but consisting of very different features, sometimes generated within different scientific disciplines. In the next section, we further discuss best practice and provide advice for interpretation of CLARITY results.

# 3. Discussion

## 3.1. Best practice

CLARITY is simple to use and has very few moving parts. There are, however, a few critical decisions:

1. designing the input (dis)similarity $Y$, for which the user should take care;
2. choosing a structure representation $A$, for which the defaults provide good performance;
3. whether to use null hypothesis statistical testing, to identify statistically significant deviations; and
4. how to interpret the results.

### 3.1.1. How to choose a (dis)similarity

CLARITY can work with rich input data $Y$. However, the simpler the measure, the more reliable the inference will be. Therefore, if you can compute a regular covariance or Euclidean distance, this will be more interpretable.

The *self-similarity* should be correctly quantified. Broadly, this should mean, 'if we removed any excess similarity that is unique to the subject, how similar would it be to itself?' We implemented a 'diagonal removal' for this purpose, which sets the diagonal to the next-highest value. This is generally recommended, and was applied in the simulations, Methylation and Culture examples. We also provide a (slower) iterative model in which $\hat{Y} = AXA^T + D$ is iteratively solved for a diagonal matrix $D$. This was used in the Linguistic example. We did not find any qualitative difference between the methods. Sensitivity analysis will confirm for users whether working with the raw similarity, or diagonal-corrected similarity, impacts inference.

Whether to *centre* the data is an important decision. Centring changes whether any difference in mean is used in the comparison, and is therefore a modelling decision. We only recommend centring if the mean is known *a priori* to be unimportant or misleading. We only centred the Culture/Economics data, which required feature standardization. Again, we recommend either careful justification of the choice to centre, or sensitivity analysis to confirm it does not matter. Conversely, *scaling* will typically not affect the inference because any change can be accounted for in $X$.

We note a clear distinction between two classes of data. The first, represented in the Language example, is where every subject is distinct, or the total amount of data is 'small'. In this case, self-

similarity correction is important as it affects $A$ closely. Further, persistences have relatively small ranges of $k$ to persist over, and therefore inference is subtle. Conversely, the second class has a 'large' number of subjects, such as the Methylation example. In this case the effect of the diagonal becomes negligible as the data grows, as can be verified in a sensitivity analysis.

### 3.1.2. How to choose the structure

In general, the unconstrained SVD-based structure is recommended; it scales to large data, and provides the best overall fit. Although the representation of structure $A$ is a linear embedding using only the SVD, it is flexible in prediction of $Y_2$ due to using $k^2$ free parameters in $X_2$.

We only recommend the mixture model when the data are small, and suspected to lie very close to some interpretable model such as a tree containing mixtures. The residuals should be close to identical between the methods, with the important difference being the interpretation of $A$. We note that unlike in many mixture model inference methods, there is nothing in our loss function that encourages $A$ to be close to the boundaries (i.e. some $A_{ij}$ are close to 0 or 1), except the initial conditions. This feature should therefore be used with further validation, as part of the hypothesis generation process.

### 3.1.3. Whether to perform null hypothesis statistical testing

CLARITY is in general a hypothesis generating tool and much of the value can be obtained without any use of statistical testing. For many datasets, statistical power will dominate (such as the Culture example) and many persistences are expected to be significant. We then care about practical significance, and hypothesis testing is somewhat spurious. Similarly, in the Methylation example, clear persistent residuals were observed and these require no testing to confirm anomalies.

Conversely, especially with small datasets, statistical power is limiting (such as the Language example), in which case we wish to check that a given persistence is significant. The statistical test that we provide compares $Y_2$ with the distribution of left-out data from $Y_1$. It does so by first mean centring and scaling each dataset *to the reference*, and then applying the standard CLARITY correction of learning-independent Relationships. This encodes an implicit assumption that the signal and noise are of the same scale, and therefore should only be used if this is plausible.

Null hypothesis testing is a non-trivial process to implement because it requires that the user is able to either create independent features, or able to provide a set of bootstrapped samples generated on pseudo-independent samples, for both the reference and the test dataset. This comes with a computational cost, as of the order of 200 pseudo-independent replicates are required to confidently reject at the 0.05 level.

We emphasize that the principal use case is the identification of large persistent residuals which CLARITY will only create where structure has changed, and does not require testing.

### 3.1.4. How to interpret persistences and residuals

We have noted that high residuals at a single complexity $k$ might capture only a small change in the importance of a particular relationship, such as two clusters getting further apart. Conversely, CLARITY is easiest to interpret when residuals are large and persist across a range of complexities. This will typically highlight structural anomalies in certain subjects.

There are two phenomena that require care. The first is that the sums of squared residuals are not monotonic for a particular subject (but are overall). For example, figure 1c(iii) shows residuals increasing for subject $i$ up to complexity 3, and then decreasing; similarly, subject $l$ has high residual at complexity 12–13. This occurs when the change to the structure fits other, nearby but different, subjects which can 'drag' that subject away from its target.

The second important phenomenon is that residuals need not be largest in the subject that has changed structure. Because the inferred relationship $X_2$ is chosen to minimize the total squared residuals, the model may make nearby subjects have the highest residuals. The important thing to note is that this still creates some residual excess in the target subject, which can be identified by looking at the matrix of residuals (e.g. figure 8c).

## 3.2. Conclusion

CLARITY can be applied to any pair of datasets in which subjects are matched and so can be used in a wide variety of situations. We demonstrated this in very different fields: epigenetics, linguistics and

bridging sociology and economics. In these examples, we have recovered differences, supported by well-documented evidence and generated new hypotheses.

The software requires very little technical knowledge to employ, there are no tuning parameters, and the output can be presented in a simple, interpretable chart we called the residual persistence. This identifies the clusters and subjects that are poorly predicted, and allows interpretation of which other clusters they may share additional structure with. We suggest that the same approach may yield valuable insights when applied to other fields of interest, and that the results will generate hypotheses for further investigation through the application of additional, statistically robust, methods.

CLARITY is fast, and, for prediction, is limited only by the cost of computing a singular value decomposition. We showed via simulation that the SVD approach is representing the structure in the data very similarly to a mixture model, for which we presented a novel algorithm based on a multiplicative update rule. The mixture model correctly identifies hierarchical structure, clusters and mixtures when these are present in the data and so permits the interrogation of why a particular prediction may have been made. We were unable to find tools that were able to perform an analogous structural comparison and, therefore, have not performed statistical recall and efficiency benchmarking. While we could have run the models listed in the introduction, they have different null hypotheses and purposes. Some provide qualitatively different information to CLARITY, while others test for equality of the similarities, which is an implausible null hypothesis for our examples. While CLARITY is currently performing a unique function in terms of information extraction from complex data, we anticipate that the problem may be addressed in other ways, and that the insights which can be automatically extracted can be extended.

# 4. Methods

## 4.1. Notation

The notation we use is largely standard. Matrices are denoted by upper case letters. The set of all $d \times k$ matrices with real entries is denoted by $\mathbb{R}^{d \times k}$. If $Y \in \mathbb{R}^{d \times k}$ is a matrix, its $(i, j)$-entry is denoted by $Y_{ij}$. The quantity $\|Y\|_F$ denotes the Frobenius norm of $Y$, i.e.

$$\|Y\|_F := \left( \sum_{i=1}^{d} \sum_{j=1}^{k} Y_{ij}^2 \right)^{1/2}.$$

## 4.2. Structural representation

In general, CLARITY can be applied to any pairs of matrices. In practice, the utility of the subsequent results does depend on how the matrix was constructed and the 'best practice' input is likely to be a distance matrix corresponding to a reasonable metric, such as Euclidean. In limited experimentation, asymmetry does not appear to be a major problem but the class of matrices we can prove that CLARITY is sensible for is the following.

A dissimilarity matrix is defined to be any symmetric matrix $Y \in \mathbb{R}^{d \times d}$ of full rank consisting of non-negative entries. Let $Y_1$ and $Y_2$ be a pair of dissimilarity matrices in $\mathbb{R}^{d \times d}$. For each natural number $k \leq d$, we initially seek matrices $A_k \in \mathbb{R}^{d \times k}$ and $X_1^{(k)} \in \mathbb{R}^{k \times k}$ such that the quantity

$$\|Y_1 - A_k X_1^{(k)} A_k^T\|_F$$

is minimized. Note that the squared error discussed in the text is the squared Frobenius norm and is minimized at the same $A$ and $X$.

The product $A_k X_1^{(k)} A_k^T$ is to be viewed as the best rank $k$ approximation of $Y_1$ in Frobenius norm subject to whatever constraints may be placed on both $A_k$ and $X_1^{(k)}$ and it affords a structural reduction of $Y_1$ at dimension $k$ as motivated by the following proposition.

**Proposition 4.1.** *Let $Y \in \mathbb{R}^{d \times d}$ be a dissimilarity matrix and let $(A, X)$ be a pair of matrices such that*

$$\|Y - AXA^T\|_F$$

*is minimized, where A has full column rank. Then*

$$X = P_A Y P_A,$$

*where $P_A$ denotes the orthogonal projection operator onto* $\mathrm{im}(A)$.

   *Proof.*  Define the objective function

$$\mathcal{L}(A, X) := \frac{1}{2}\|Y - AXA^T\|_F^2.$$

Taking matrix derivatives with respect to $X$ gives the condition

$$A^T(AXA^T - Y)A = 0,$$

at a critical point $(A, X)$. If $A$ has full column rank, the matrix $A^TA$ is invertible and it is possible to solve for $X$ by

$$X = A^+ Y (A^+)^T,$$

where $A^+ := (A^TA)^{-1}A^T$ is the generalized (Moore–Penrose) inverse of $A$. Then

$$AXA^T = AA^+ Y(AA^+)^T = P_A Y P_A.$$

$\blacksquare$

   Given the above structural reduction of $Y_1$, we seek to find the extent to which it is capable of predicting the matrix $Y_2$. To this end, we find a matrix $X_2^{(k)}$ such that

$$\|Y_2 - A_k X_2^{(k)} A_k^T\|_F$$

is minimized and we examine both the residual matrix

$$Y_2 - A_k X_2^{(k)} A_k^T,$$

and element-wise norms of it.

   If $A_k$ has full column rank, the argument in proposition 4.1 gives that $X_2^{(k)} = P_{A_k} Y_2 P_{A_k}$ where $P_{A_k}$ denotes the orthogonal projection onto $\mathrm{im}(P_{A_k})$.

## 4.3. Learning structure

We consider two methods that differ only in the manner in which the initial optimization problem stated above is solved. Our *SVD model* uses singular value decomposition to solve analytically for $A_k$ and $X_1^{(k)}$, and it is possible to do this precisely because these matrices are assumed to be unconstrained. Our *Mixture model* constrains the matrix $A_k$ to have rows taken from a probability simplex (but does not constrain $X_1^{(k)}$), and an optimum is obtained numerically via an iterative procedure.

### 4.3.1. SVD-based solution

Suppose that we have singular value decomposition

$$Y_1 = \sum_{j=1}^{d} \sigma_j u_j v_j^T,$$

where $\sigma_j$ denotes the $j$-largest singular value of $Y_1$. The matrix product $A_k X_1^{(k)} A_k^T$ can have rank at most $k$, and by the Eckart–Young theorem [59],

$$\min_{Y':\,\mathrm{rk}(Y')\leq k} \|Y - Y'\|_F = \|Y - \hat{Y}_{1,k}\|,$$

where $\hat{Y}_{1,k}$ is defined to be the truncation of the SVD of $Y_1$ to its top $k$ singular values, i.e.

$$\hat{Y}_{1,k} := \sum_{j=1}^{k} \sigma_j u_j v_j^T.$$

We set $A_k = [u_1|u_2|\ldots|u_k]$. The matrix $X_1^{(k)}$ is then the top left-hand $k \times k$ block of $\Sigma_1$, and

$$X_2^{(k)} = A_k^T Y_2 A_k.$$

## 4.3.2. Solution under a simplicial constraint

We assume that the entries of $A_k$ are non-negative and that the rows of $A_k$ sum to 1. This constraint is motivated by mixture modelling. A solution is sought via an iterative gradient descent procedure using multiplicative update rules based on the approach of Lee & Seung [60].

Specifically, we derive a multiplicative update rule for $A$ given $X$ and $Y_1$ and then solve for $X$ given $A$ and $Y_1$. These two steps are applied to convergence. This particular model does not appear to have been solved previously in the literature, and this solution is relatively efficient.

Given $X$ and $Y_1$, consider the objective function

$$\mathcal{L}(A) := \frac{1}{2} \| Y_1 - AXA^T \|_F^2.$$

If $A_t$ is the current estimate of $A$ at iteration $t$, taking matrix derivatives of $\mathcal{L}(A)$ leads to the update rule

$$(A_{t+1})_{ij} \leftarrow (A_t)_{ij} \frac{(N_t^A)_{ij}}{(D_t^A)_{ij}},$$

where

$$N_t^A = Y_1^T A_t X_t + Y_1 A_t X_t^T,$$

and

$$D_t^A = A_t X_t A_t^T A_t X_t^T + A_t X_t^T A_t^T A_t X_t,$$

and $X_t$ denotes the estimate of $X$ at the $t$th iteration. If $A_{t+1}$ has full column rank, then we solve for $X_{t+1}$ by use of the generalized inverse; i.e.

$$X_{t+1} = A_{t+1}^+ Y_1 (A_{t+1}^+)^T.$$

If $A_{t+1}$ does not have full column rank, a multiplicative update rule is used to update $X_t$ derived analogously, i.e.

$$(X_{t+1})_{ij} = (X_t)_{ij} \frac{(N_t^X)_{ij}}{(D_t^X)_{ij}},$$

where

$$N_t^X := A_{t+1}^T Y_1 A_{t+1},$$

and

$$D_t^X := A_{t+1}^T A_{t+1} X_t A_{t+1}^T A_{t+1}.$$

Empirically, the row-sums are approximately stable in this algorithm, but it does *not* guarantee that the rows sum to 1. Therefore, at each iteration we renormalize the rows to enforce this property. The row sums are not in general identifiable. In practice, disabling this normalization does not allow the row sums to drift significantly, except in cases where the model is a very poor approximation to the data. Poor model fit may cause the algorithm to terminate because it cannot find the local optima.

The following algorithm describes this rule, using $\circ$ to denote the entry-wise product of two matrices.

---

**Algorithm 1.**

Inputs: Data $Y$, initial value of $A_0$; maximum number of iterations $t_{\max}$

**for** $t = 1 \ldots t_{\max}$ **do**

 $A_t = \text{NormalizeRows}(A_{t-1} \circ (N_{t-1}^A / D_{t-1}^A))$

 **if** $A$ has full column rank **then**

 $X_t = A_{t-1}^+ Y_1 (A_{t-1}^+)^T$

 **else**

 $X_t = X_{t-1} \circ (N_{t-1}^X / D_{t-1}^X)$

 **end if**

 **if** $\| A_t - A_{t-1} \| < \delta$ **then** Break

**end for**

Outputs: Estimates $A = A_t$ and $X = X_y$

---

## 4.4. Statistical significance

For simple datasets consisting of $d$ subjects about which we observe $L$ features, significance is measured using a statistical resampling procedure implemented in the CLARITY package. More complex datasets where similarities are computed in a complex way, and not read straightforwardly off matches between features—for example, as for our linguistic data—can still be quantified via resampling. In such cases, the data are bootstrapped externally and provided to the software as a set of matrices. In this procedure, we sample $L/2$ of the $L$ features (columns) of the data $D_1$, which is a $d \times L$ matrix. We then compute a 'sampled reference' (dis)similarity matrix, and from the remaining $L/2$, a 'sampled target' (dis)similarity matrix. We then replicate the downsampling procedure on the target data and obtain a (dis)similarity matrix. We then mean centre and scale both sampled target and downsampled original target matrices into the sampled reference matrix, and evaluate test statistics $f$ (squared residuals and persistences). This is repeated $n_{bs}$ times.

We compute a regularized empirical $p$-value $p(f(Y_2)|f(Y_1)) = (1/(1+n_{bs}))(1 + \sum_{i=1}^{n_{bs}} \mathbb{I}(f(Y_2) \geq f(Y_1))$, formed from the probability that a sample from the true target is as great or greater than the resampled reference. This procedure is necessary because bootstrap resampling [61] is not straightforwardly valid for similarity matrices.

Whilst this procedure correctly estimates which structures of $Y_2$ are not predicted by $Y_1$, it does not distinguish between structures that are generated by signal versus noise. Because we are not interested in predicting noise, we further need to detect it. Estimating values of $k$ associated with structure is straightforward by simple cross-validation, because we have already constructed many random resamples of the data. We can therefore predict fold-2 of $Y_2$ from a CLARITY model learned in fold-1 of $Y_2$ and estimate $\hat{k}$ from the minimum cross-validation error.

Because $\hat{k}$ is a point estimate subject to variation, we obtain $n_{bs}$ samples $\hat{k}_i$ and implement a soft threshold, the 'probability that complexity $k$ is describing structure' $p(k) = (1/n_{bs}) \sum_{i=1}^{n_{bs}} \mathbb{I}(\hat{k}_i \geq k)$, i.e. the proportion of bootstrap samples that have an estimate at least as large as $k$. We then report the complete CLARITY $p$-value $p_c$, quantifying the 'probability of observing a test-statistic for $Y_2$, this extreme or more so, under the null hypothesis that $Y_2$ is a mean scaled (dimension $k$ rotation, translation and scaled) version of $Y_1$ within which Complexity $k$ describes Structure'. This is:

$$p_c = 1 - \left( (1 - p(k)) p(f(Y_2)|f(Y_1)) \right),$$

which is close to 0 only if both $p(k)$ is close to 1 and $p(f(Y_2)|f(Y_1))$ is close to 0.

Because the $p$-values are highly correlated, and there is a multiple testing burden, the $p$-values should not be used to test for the presence of any difference in structure between $Y_1$ and $Y_2$. In particular, it can be that $Y_1$ and $Y_2$ are substantially different, but this does not result in any particular cell in the persistence diagram having a $p$-value at the appropriate multiple-testing level. In other words, on the level of $Y_1$ and $Y_2$ viewed globally, testing individual residuals has very low power; testing individual persistences has low power. A more powerful test would be for $f$ being the Frobenius norm of the whole matrix, but this has limited scientific value as it discards the scientific significance of the results.

We provide a formal statistical significance procedure to add weight to the identification of scientifically significant results, but emphasize that these are different concepts.

## 4.5. Methylation/expression simulated data

For figure 4, we use the simulation data from [36] (electronic supplementary material, figure S3) which is based on real methylation and expression patterns.

First each locus is assigned a methylation class, 300 are 'hypo'methylated (low) and 700 are 'hyper'methylated (high). They are then given a mean methylation, $\mu_i \sim U(0.1, 0.4)$ and $\mu_i \sim U(0.55, 0.85)$ for hypo/hyper, respectively. Then 'Tumour' cases are assigned methylation $m_{ij} \sim U(\mu_i - \mu_i/2, \mu_i + \mu_i/2)$ or $m_{ij} \sim U(\mu_i - (1 - \mu_i)/2, \mu_i + (1 - \mu_i)/2)$ for hypo/hyper. Control cases follow the same distribution but shift towards the mean by replacing $\mu_i$ with $\mu_i + 0.2$ (if $\mu_i < 0.3$), or $\mu_i - 0.2$ (if $\mu_i > 0.7$) or with equal probability of positive or negative shift $\pm 0.2$ otherwise.

To create Expression data, the procedure generates $\sigma_i \sim U(0.5, 0.9)$ for each locus and then sets $e'_{ij} \sim N(\mu_{ij}, \sigma_i^2)$. The reported expression $e_{ij}$ are then centred and scaled.

We then chose two segments of 10 loci and moved the two classes towards each other; the top anomaly loci have 'tumour' expression altered, and the bottom have 'control' altered, by setting $e_{ij}^{anomaly} = -2e_{ij}$. This simulates loci that behave differently in Methylation data to in Expression data. These parameters induce an average $-5\%$ correlation between methylation and expression, i.e. the association is weak.

## 4.6. Simulation details

For §2.4, we generate a coalescent tree $\mathcal{T}_1$ using 'rcoal' from the package 'ape' [62] for R [63]. The 'true' $A$ is a vector of zeroes except for the cluster membership $k$ of $i$, for which $A_{ik} = 1$. We then simulate a matrix $D_0$ consisting of $k$ rows (clusters) and $L$ columns (features) by allowing features to drift in a correlated manner under a random-walk model using the function 'rTraitCont' from the package 'ape'. This generates a 'true' $X = \text{Dist}(D_0)$. To generate a feature $d$ for a sample with mixture $a$, we simulate features $d\, N(a^T D_o, \sigma_0^2)$, from which we can compute $Y = \text{Dist}(D)$.

In *Scenario A*, we make $\mathcal{T}_2$ into a non-ultrametric tree by randomly perturbing the branch lengths of $\mathcal{T}_1$ by multiplying each by a $U(0.1, 2)$ variable. We generate $Y^{(2)}$ as above from $\mathcal{T}_2$.

In *Scenario B*, we make $\mathcal{T}_2$ as in Scenario A. Then one additional *mixture edge* is added at random. This is done by choosing a tip of the tree $i$, choosing a second tip $j$ at least the median distance from the first tip, and setting $A[, i] \leftarrow (1 - \beta)A[, j]$ and $A[, j] = A[, j] + \beta A[, i]$. This edge affects a proportion $r$ of the subjects in cluster $i$. If $r = 0$ or $r = 1$ this becomes a relationship change rather than a structural change, because all of the samples in the cluster adopt a new relationship with the remaining clusters (though the relationship is no longer a tree). We use $r = 0.5$ throughout.

## 4.7. Language example details

We compute **phonetic similarities** using the *information-weighted distance with sound correspondences* (IWDSC) method [44]. First, we estimate global sound similarity scores, based on the whole NorthEuraLex 0.9 dataset with 107 languages. This provides us with an idea of which sounds in the data generally tend to be close. Implicitly, the employed inference method makes those sounds close that appear in words that are probably historically related. In other words, global sound similarities are not directly about articulatory or auditory similarities (i.e. how humans produce and perceive different sounds), but rather estimate 'historical similarity', thus implicitly tracking processes of language change.

After obtaining global sound similarities, we compute local sound similarity scores for each language–language pair in our 36-language Indo-European subset of NorthEuraLex. This works similarly to global similarity scores, but now only taking into account data from those two languages. Both global and local sound similarity scores are based on mutual information. In particular, the local scores declare such sounds similar which are highly predictable from the sounds in the word expressing the same meaning in the other language.

To obtain overall language–language Phonetic scores, we first build word-to-word similarity scores based on sound-to-sound similarity scores. Crucially, we discount the weight of the sounds in highly regular parts of words, e.g. the infinitive ending in German verbs such as geb*en* 'to give' and leb*en* 'to live' [64]. This way, we discount the regular grammatical elements: they carry information about the grammar of a language, but not about its individual words. We also normalize by word length. To get aggregate language-to-language similarities out of word-to-word similarities, we simply average.

Language-to-language **lexical similarity** is defined as cognate overlap: the share of words in the relevant two languages that were inferred to have the same ancestral word. We produce automatic cognacy judgements by applying  unweighted pair group method with arithmetic mean (UPGMA) clustering to the word-to-word phonetic similarity scores within each meaning, a method shown to currently produce state-of-the-art automatic cognacy judgements [44].

Both Phonetic and Lexical similarities that we compute are based on word-to-word phonetic similarity scores. The cognate clustering step that takes us from word-to-word similarities to cognate overlap aims to uncover, automatically, information about the word-replacement change. Phonetic and Lexical information is bound to be **highly correlated**. First, the change of two types occurs in the same communities subject to the same historical processes. For example, both Phonetic and Lexical change accumulate with time, so two speech communities that split earlier will be more dissimilar on both Phonetic and Lexical change than two speech communities with a later split, other things being equal. Second, when two languages retain a common ancestral word, simply by virtue of stemming from the same proto-word, the two modern words are going to be more phonetically similar than two randomly selected phonetic sequences from the two languages. Thus higher levels of true cognate overlap will lead to higher levels of phonetic similarities.

Finally, in addition to these two real-world drivers of correlation, in our computational analysis we infer lexical overlap based on low-level phonetic similarity. It is a common and effective practice in computational historical linguistics, and only slightly inferior to expert-coded information for at least some types of practical inference [65]. But we do expect to miss some true cognates that changed phonetically so much as to be not statistically identifiable from the raw data without additional expert

knowledge. This makes our estimated dissimilarity matrices for Phonetic and Lexical still more correlated than the corresponding ground-truth matrices would be. This makes it all the more striking that despite a strong correlation between Phonetic and Lexical, stemming from both natural and analysis-induced sources, we find a robust and convincing effect of mismatch using CLARITY.

To assess **significance**, we use the method described in §4.4, dividing the data into two halves by meaning, and computing independently a similarity matrix from each half. When doing that, we always use the same global similarity scores, which represent the properties of a much larger sample of 107 languages, taken as a proxy for languages of the world in general.

We use **cross-validation** to assess whether CLARITY decomposition at higher complexities $k$ still captures signal rather than noise. For the pairs of matrices based on two halves of the data, we predict one based on the other (so Phonetic from Phonetic and Lexical from Lexical). This shows (electronic supplementary material, figure S3) that even at the highest $k$ we do not have overfitting to noise. Therefore, the full range of $k$ in the main analysis is interpretable.

The CLARITY set-up in this example implies that the diagonal values in our similarity matrices might not be amenable to successful modelling. In the main text above, we report the results where we discount the diagonal when doing CLARITY decomposition. We checked whether the pattern we found depended on this choice. Electronic supplementary material, figure S4 shows CLARITY persistences with no special treatment for the diagonal, and those are basically the same as those in figure 7. Similarly, we checked (electronic supplementary material, figure S5) that lowering the significance threshold from 0.05 to 0.01 does not change the result. Finally, in electronic supplementary material, figures S6 and S7), we checked that the image representation of persistence did not affect our inference, and that resampling the persistence curves for the matrices based on half the data, used for significance testing. Taken together, these further checks convince us that our reported result is real and not spurious.

## 4.8. Mathematical validity of structural comparison

The matrices $Y_1$ and $Y_2$ are typically observed with non-independent noise, and so there is a need for the various quantities of interest to be stable under perturbation—that is, that a small change in the data does not result in a large change to the inference. The following result describes the stability of the residual matrix under perturbation of $Y_1$ and $Y_2$ for the SVD-based solution.

**Theorem 4.2.** *Let* $Y_1, Y_2, Y'_1, Y'_2 \in \mathbb{R}^d$ *be symmetric matrices such that* $\|Y_2 - Y'_2\|_F$, $\|Y_1 - Y'_1\|_F \leq \epsilon$. *Suppose that we have singular value decompositions* $Y_1 = U_1 \Sigma_1 V_1^T$ *and* $Y'_1 = U'_1 \Sigma'_1 V'^T_1$. *Let* $A_k$ *and* $A'_k$ *be the matrices obtained by taking the first* $k$ *columns of* $U_1$ *and* $U_1'$, *respectively.*

1. *If* $X_2^{(k)} = A_k^+ Y_2 (A_k^+)^T$ *with* $X_2^{'(k)}$ *defined analogously for* $Y'_2$ *then*

$$\|Y_2 - A_k X_2^{(k)} A_k^T\|_F \leq \|Y'_2 - A_k X_2^{'(k)} A_k^T\|_F + 2\epsilon.$$

2. *Suppose that* $Y_1$ *has eigenvalues* $\lambda_1 \geq \lambda_2 \geq \ldots \geq \lambda_d$ *and let* $\delta_k := \lambda_k - \lambda_{k+1}$ *for each natural number* $k < d$. *Then*

$$\|Y_2 - A_k X_2^{(k)} A_k^T\|_F \leq \|Y_2 - A'_k X_2^{(k)} A_k^{'T}\|_F + \frac{2^{5/2}\epsilon}{\delta_k}.$$

The proof of theorem 4.2 may be found in appendix A. Theorem 4.2 can be used for statistical purposes as follows. If $Y'_1$ and $Y'_2$ are sampled matrices that are believed to be close to their population counterparts $Y_1$, $Y_2$ (for example, when dealing with covariances), then given suitably sized eigengaps $\delta_k$ and $\delta_k'$ for $Y_1$ and $Y_1'$, respectively, the Frobenius norm of the estimated residual matrix is close to that of the true residual matrix. Specifically, simple manipulation of the inequalities established in theorem 4.2 leads to the deviation inequality

$$\left| \|Y_2 - A_k X_2^{(k)} A_k^T\|_F - \|Y'_2 - A'_k X_2^{'(k)} A_k^{'T}\|_F \right| \leq \frac{2 + 2^{5/2}}{\min(\delta_k, \delta'_k)} \epsilon$$

where $\|Y_2 - Y'_2\|_F$, $\|Y_1 - Y'_1\|_F \leq \epsilon$.

Data accessibility. Data and relevant code (R packages *CLARITY* and *CLARITYsim*) for this research work are stored in GitHub: github.com/danjlawson/CLARITY and have been archived within the Zenodo repository: https://doi.org/10.5281/zenodo.5172063.
Competing interests. We declare we have no competing interests.

Funding. D.J.L. is funded by the Wellcome Trust and Royal Society Sir Henry Dale Fellowship, grant no. WT104125MA. D.J.L. and P.E. are supported by the OCSEAN grant funded by the EU Research Executive Agency (Horizon 2020 MSCA RISE 2019 number 873207). J.D. has been supported by the German Research Foundation (DFG) under FOR 2237 'Words, Bones, Genes, Tools' and by the European Research Council (ERC) under the Horizon 2020 research and innovation programme (CrossLingference, grant agreement no. 834050, I.Y. has been supported by the German Research Foundation (DFG) under Emmy-Noether-NWG 391377018 and under FOR 2237 'Words, Bones, Genes, Tools'.

# Appendix A. Proofs

## A.1. Notation

In addition to the notation already introduced, if $A$ is a matrix, its spectral norm is denoted by $\|A\|_2$. The singular values $\sigma_1(A) \geq \sigma_2(A)\ldots$ of $A$ are listed in non-increasing order, and so $\|A\|_2 = \sigma_1(A)$. If $v$ is a vector, its Euclidean norm is denoted by $\|v\|$. If $A$ is a matrix, its vectorization (the vector obtained by stacking the columns of $A$) is denoted by $Vec(A)$.

## A.2. Preliminary facts

Recall that for any matrices $A$, $B$ and $C$ where the product $ABC$ exists, we have the identity $Vec(ABC) = (C^T \otimes A)Vec(B)$ where $\otimes$ denotes the Kronecker product of two matrices. This identity is useful in what follows.

If $V$, $V'$ are $d \times k$ matrices with orthonormal columns, we have a vector $(\cos^{-1}(\sigma_1), \ldots, \cos^{-1}(\sigma_k))^T$ of principal angles, where the $\sigma_j$ are the singular values of $V'^T V$. Let $\Theta(V', V)$ denote the $r \times r$ diagonal matrix with the $j$th diagonal entry given by the $j$th principal angle. The matrices $\sin \Theta(V', V)$ and $\cos \Theta(V', V)$ are defined entry-wise. The perturbation bounds established rely on the following variant of the Davis–Kahan theorem [66].

**Theorem A.1.** *Let $Y$, $Y' \in \mathbb{R}^{d \times d}$ be symmetric matrices with eigenvalues $\lambda_1 \geq \ldots \geq \lambda_d$ and $\lambda'_1 \geq \ldots \geq \lambda'_d$, respectively. Fix $1 \leq r \leq s \leq d$ and suppose that $\delta_{r,s} := \min (\lambda_{r-1} - \lambda_r, \lambda_s - \lambda_{s+1}) > 0$, where $\lambda_0 := \infty$ and $\lambda_{d+1} := -\infty$. Put $p = s - r + 1$ and define $V := [v_r | v_{r+1} | \ldots | v_s]$, $V' := [v'_r | v'_{r+1} | \ldots | v'_s]$, both with orthonormal columns, satisfying $Yv_j = \lambda_j v_j$ and $Y'v'_j = \lambda'_j v'_j$ for each $j = r, r+1, \ldots, s$. Then*

$$\| \sin \Theta(V', V) \|_F \leq \frac{2\min(p^{1/2}\|Y - Y'\|_2, \|Y - Y'\|_F)}{\delta_{r,s}}.$$

## A.3. Proof of theorem 4.2

1.  Let $P_k = P_{A_k}$. Then

$$\begin{aligned}
\|A_k(X'_2 - X_2)A_k^T\|_F &= \|P_k(Y'_2 - Y_2)P_k\|_F \\
&= \|(P_k \otimes P_k)Vec(Y'_2 - Y_2)\| \\
&\leq \|P_k \otimes P_k\|_2 \|Y'_2 - Y_2\|_F \\
&= \epsilon,
\end{aligned}$$

    and the claim follows by the triangle inequality.

2.  Let $P'_k = P_{A'_k}$. Then,

$$\begin{aligned}
\|P'_k Y_2 P'_k - P_k Y_2 P_k\|_F &\leq \|(P'_k - P_k)Y_2 P'_k\|_F + \|P_k Y_2(P'_k - P_k)\|_F \\
&\leq \|(P'_k \otimes (P'_k - P_k))Vec(Y_2)\| + \|(P'_k - P_k) \otimes P_k)Vec(Y_2)\| \\
&\leq 2\|P'_k - P_k\|_2 \|Y_2\|_F.
\end{aligned}$$

    Moreover,

$$\begin{aligned}
\|P'_k - P_k\|_2^2 &\leq \|P'_k - P_k\|_F^2 \\
&= \|P'_k\|_F^2 + \|P_k\|_F^2 - 2Tr(P_k P'_k) \\
&= 2(k - \| \cos \Theta(U'_k, U_k)\|_F^2) \\
&= 2\| \sin \Theta(U'_k, U_k)\|_F^2 \\
&\leq 8\frac{\|Y_1 - Y'_1\|_F^2}{\delta_k^2},
\end{aligned}$$

    where theorem A.1 has been used to obtain the last inequality. The claim follows by the triangle inequality.

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
