## [Peer Review File · Royal Society Open Science]

Review History

RSOS-202182.R0 (Original submission)

Review form: Reviewer 1

Is the manuscript scientifically sound in its present form?

Yes

Are the interpretations and conclusions justified by the results?

Yes

Is the language acceptable?

No

Do you have any ethical concerns with this paper?

No

Have you any concerns about statistical analyses in this paper?

No

Recommendation?

Major revision is needed (please make suggestions in comments)

Comments to the Author(s)

Overall, I found the paper very interesting and the method potentially very useful, especially for datasets that come from several scientific disciplines and/or are generated using different methodologies.

However, I do think the paper needs some work before it can be published in Open Science, especially in what concerns the way it is presented given its potential audience.

More precisely, I found the paper **very hard to read and digest**, on the one hand, and not uniform in terms of writing (I have the impression that different parts are clearly written by different people having different audiences in mind without a final "harmonization" of the manuscript). This is particularly troubling as it might impact the adoption of the method across disciplines, on the one hand, and will certainly muddle its interpretation, on the other. I hope that this rather general comment might become clearer later on...

Now, to more specific issues:

- number of examples in the paper: the abstract says 3, the introduction says 2; while I know what this means **after** reading the paper, it does not inspire trust

- Figure 1: this is supposed to "ease people" into the paper and summarize what the method is about but I found it very confusing and hard to understand. For example, in panel (b), subpanel (i) it is unclear what the color patches next to the tree are and again if they correspond to those in subpanel (ii). The text says "Data samples are generated as observed with features, belonging to clusters under a 'true structure', all hierarchical clusterings in the data." but this is hard to parse and hard to understand given the amount of information given so far (it becomes clear after reading the paper, but it might be too late by then). Minor things: "For" -> "for" and I guess small k and big K are the same thing?

- p5, l124: "relationship between structures" - this is confusing here and should much better explained

- the linguistic example: while I do have some questions about it (more later), the main gripe I have here is that, as opposed to the cultural one, there is no attempt at analyzing particular languages so as the reader can really see the usefulness of the method besides of "look, differences that you wouldn't see with the naked eye!" Personally, I find it very intriguing that the two directions of prediction look relatively similar but yet so different, and I would have liked some attempted comment on why, for example, the Romance languages (except Romanian) are so deviant and the Germanic ones (except English) when precisely Romanian and English should be relatively "weird" given their history of contact with other languages.

- the cultural example: first, why would one predict culture from economy? I would have naively expected the exact opposite direction. However, I found this example very well explained and it actually helped me get my head around the outputs of the method! Nevertheless, I was a bit surprised by the interpretation of Andorra (why the Middle East???) but, on the other hand, it was good to see Poland singled out (but I would have naively expected the Czech Republic and Hungary and even East Germany -- if it was there -- to show a similar deviation).

- discussion: I have to say that I still do not know how to interpret the deviations, which makes it even more frustrating given that this is said to be a major advantage of the method. Moreover, even if I think I know why, the constant flip in the manuscript between "it's about similarities" and "it's about differences" makes the reader dizzy.

- methods: on page 3:45-47 it is said "Unlike distances, dissimilarities need not be symmetric, nor satisfy other useful properties such as the triangle inequality." but then on p13:343 it is said "a dissimilarity matrix is defined to be any symmetric matrix"

- methods: p14:367-368: the clear statement and description of the *2* methods should come much earlier in the paper -- I was really confused as I thought there were two methods but then I wasn't fully sure

- methods: statistical significance: "However, on the level of subjects, we expect the formally significant results to be scientifically significant" -- I'll be a complete bore, but I am afraid there's quite little overlap between statistical and scientific significance, and what such p-values can do, at best, is signal places where it *might* be worth insisting (on the other hand, for such a method, also places with clearly n.s. p-values might be as equally worth looking at). So, just as a naive suggestion: maybe better to simply show somehow the range/distribution of the resampled values versus the empirical one (as, e.g., sort of stacked series of intervals or such) and let the people decide?

- p18:457: I think something is missing between d and N (maybe "~" ?)

- methods: language example: I am afraid I am not familiar with the particular method developed by Dellert and colleagues, but, naively, it strikes me that maybe one can also use expert cognacy judgments available for the basic vocabulary e.g., in IE, Uralic, Bantu... languages and see how that looks like? It is not that I do not trust the method itself, but I feel relatively unhappy to use it as an *example* for validating a *new method* -- I would have expected to use cases where everybody knows what's going on or, at least, where surprising results cannot be easily relegated to process producing the data

Thus, what I would suggest is:

1. use a completely made-up *toy example* that is easy to grasp for Figure 1 and also show, graphically, with boxes and arrows, what the method *does* and how to *interpret* its output (kind of what was done for the cultural example, maybe even use this one?)
2. clarify the main text with a non-technical audience in mind
3. for the linguistic example, maybe use a more "standard" cognacy judgment (maybe on a reduced set?) and *explain* what the output *means* given what we know about the history of those languages

Decision letter (RSOS-202182.R0)

Dear Dr Lawson

The Editors assigned to your paper RSOS-202182 "CLARITY -- Comparing heterogeneous data using dissimiLARITY" have now received comments from reviewers and would like you to revise the paper in accordance with the reviewer comments and any comments from the Editors. Please note this decision does not guarantee eventual acceptance.

Please submit your revised manuscript and required files (see below) no later than 21 days from today's (ie 08-Jul-2021) date. Note: the ScholarOne system will 'lock' if submission of the revision is attempted 21 or more days after the deadline. If you do not think you will be able to meet this deadline please contact the editorial office immediately.

on behalf of Professor Joshua Ross (Associate Editor) and Mark Chaplain (Subject Editor)
openscience@royalsociety.org

Associate Editor Comments to Author (Professor Joshua Ross):

Dear Authors,

I am in agreement with the Reviewer, that this manuscript makes a valuable contribution but that to ensure its uptake, major revisions are required.

The manuscript is heavy in terminology and concepts, and attempting to make these clearer, especially earlier in the paper, would be valued.

Main overarching requests:

1. Break down the examples further, and define explicitly for the examples, what each of the terms/concepts are and correspond to. For example, what are the subjects, what are the features, what are the structures, etc.
2. Clearly outline / provide an overview of the two methods -- how they fit in to the overall CLARITY algorithm, and how they differ.

Minor requests:

1. 'similarity' is very loose in 1.1.
2. It appears that k and K are both used; and k/K are referred to as complexity/components/features. Perhaps attempt to clarify, and be consistent in term usage.
3. Use CLARITY (and not Clarity) throughout.
4. In 2.1, 'subjects' d is introduced, without a precise definition/example.
5. In 2.1, 'structures' A_k is introduced, without a precise definition/example.
6. In 2.1, " X_1 " and " X_2 " appear, without definition. Perhaps the earlier X should be X_1 to better link to Y_1 ?
7. It feels a bit confusing, in that you state something is appropriate if they are structurally similar; but is that not what you are attempting to assess?

Best regards,
Joshua Ross.

Reviewer comments to Author:

Reviewer: 1

Comments to the Author(s)

Overall, I found the paper very interesting and the method potentially very useful, especially for datasets that come from several scientific disciplines and/or are generated using different methodologies.

However, I do think the paper needs some work before it can be published in Open Science, especially in what concerns the way it is presented given its potential audience.

More precisely, I found the paper *very hard to read and digest*, on the one hand, and not uniform in terms of writing (I have the impression that different parts are clearly written by different people having different audiences in mind without a final "harmonization" of the manuscript). This is particularly troubling as it might impact the adoption of the method across disciplines, on the one hand, and will certainly muddle its interpretation, on the other. I hope that this rather general comment might become clearer later on...

Now, to more specific issues:

- number of examples in the paper: the abstract says 3, the introduction says 2; while I know what this means *after* reading the paper, it does not inspire trust

- Figure 1: this is supposed to "ease people" into the paper and summarize what the method is about but I found it very confusing and hard to understand. For example, in panel (b), subpanel (i) it is unclear what the color patches next to the tree are and again if they correspond to those in subpanel (ii). The text says "Data samples are generated as observed with features, belonging to clusters under a 'true structure', all hierarchical clusterings in the data." but this is hard to parse and hard to understand given the amount of information given so far (it becomes clear after reading the paper, but it might be too late by then). Minor things: "For" -> "for" and I guess small k and big K are the same thing?

- p5, l124: "relationship between structures" - this is confusing here and should much better explained

- the linguistic example: while I do have some questions about it (more later), the main gripe I have here is that, as opposed to the cultural one, there is no attempt at analyzing particular languages so as the reader can really see the usefulness of the method besides of "look, differences that you wouldn't see with the naked eye!" Personally, I find it very intriguing that the two directions of prediction look relatively similar but yet so different, and I would have liked some attempted comment on why, for example, the Romance languages (except Romanian) are so deviant and the Germanic ones (except English) when precisely Romanian and English should be relatively "weird" given they history of contact with other languages.

- the cultural example: first, why would one predict culture from economy? I would have naively expected the exact opposite direction. However, I found this example very well explained and it actually helped me get my head around the outputs of the method! Nevertheless, I was a bit surprised by the interpretation of Andorra (why the Middle East???) but, on the other hand, it was good to see Poland singled out (but I would have naively expected the Czech Republic and Hungary and even East Germany -- if it was there -- to show a similar deviation).

- discussion: I have to say that I still do not know how to interpret the deviations, which makes it even more frustrating given that this is said to be a major advantage of the method. Moreover, even if I think I know why, the constant flip in the manuscript between "it's about similarities" and "it's about differences" makes the reader dizzy.

- methods: on page 3:45-47 it is said "Unlike distances, dissimilarities need not be symmetric, nor satisfy other useful properties such as the triangle inequality." but then on p13:343 it is said "a dissimilarity matrix is defined to be any symmetric matrix"

- methods: p14:367-368: the clear statement and description of the *2* methods should come much earlier in the paper -- I was really confused as I thought there were two methods but then I wasn't fully sure

- methods: statistical significance: "However, on the level of subjects, we expect the formally significant results to be scientifically significant" -- I'll be a complete bore, but I am afraid there's quite little overlap between statistical and scientific significance, and what such p-values can do, at best, is signal places where it *might* be worth insisting (on the other hand, for such a method, also places with clearly n.s. p-values might be as equally worth looking at). So, just as a naive suggestion: maybe better to simply show somehow the range/distribution of the resampled values versus the empirical one (as, e.g., sort of stacked series of intervals or such) and let the people decide?

- p18:457: I think something is missing between d and N (maybe "~" ?)

- methods: language example: I am afraid I am not familiar with the particular method developed by Dellert and colleagues, but, naively, it strikes me that maybe one can also use expert cognacy judgments available for the basic vocabulary e.g., in IE, Uralic, Bantu... languages and see how that looks like? It is not that I do not trust the method itself, but I feel relatively unhappy to use it as an *example* for validating a *new method* -- I would have expected to use cases where everybody knows what's going on or, at least, where surprising results cannot be easily relegated to process producing the data

Thus, what I would suggest is:

1. use a completely made-up *toy example* that is easy to grasp for Figure 1 and also show, graphically, with boxes and arrows, what the method *does* and how to *interpret* its output (kind of what was done for the cultural example, maybe even use this one?)
2. clarify the main text with a non-technical audience in mind
3. for the linguistic example, maybe use a more "standard" cognacy judgment (maybe on a reduced set?) and *explain* what the output *means* given what we know about the history of those languages

===PREPARING YOUR MANUSCRIPT===

===PREPARING YOUR REVISION IN SCHOLARONE===

Please ensure that you include a summary of your paper at Step 2 'Type, Title, & Abstract'. This should be no more than 100 words to explain to a non-scientific audience the key findings of your

research. This will be included in a weekly highlights email circulated by the Royal Society press office to national UK, international, and scientific news outlets to promote your work.

Author's Response to Decision Letter for (RSOS-202182.R0)

See Appendix A.

RSOS-202182.R1 (Revision)

Review form: Reviewer 1

Is the manuscript scientifically sound in its present form?

Yes

Are the interpretations and conclusions justified by the results?

Yes

Is the language acceptable?

Yes

Do you have any ethical concerns with this paper?

No

Have you any concerns about statistical analyses in this paper?

No

Recommendation?

Accept with minor revision (please list in comments)

Comments to the Author(s)

I really appreciate the thoroughness of this revision and I wish to thank the authors for their efforts and for taking into accounts my comments: now the paper is definitely much clearer and easier to read and apply!

Just some small things:

- Figure 1 was missing from this version (but present in the track changes one)

- 4:48: Y_{ij}

- 5:28: "matrix whose ... row of Y ": this is unclear

- Figure 2: "a wide class of model" > models

- 5:6 "is a the Euclidean"

- 6:43: "used for as a "

- linguistic example: I think there's a problem with diacritics? I.e. German "green" and Danish "green" seem to miss their vowel (ü and ø respectively)

- 14:4 Mantel test: I am being pedantic, but Mantel's test does seem to have issues

(<https://besjournals.onlinelibrary.wiley.com/doi/10.1111/2041-210x.12018>) -- I don't think that affects your argument here, but still...

Decision letter (RSOS-202182.R1)

Dear Dr Lawson

On behalf of the Editors, we are pleased to inform you that your Manuscript RSOS-202182.R1 "CLARITY -- Comparing heterogeneous data using dissimiLARITY" has been accepted for

publication in Royal Society Open Science subject to minor revision in accordance with the referees' reports. Please find the referees' comments along with any feedback from the Editors below my signature.

Please submit your revised manuscript and required files (see below) no later than 7 days from today's (ie 25-Oct-2021) date. Note: the ScholarOne system will 'lock' if submission of the revision is attempted 7 or more days after the deadline. If you do not think you will be able to meet this deadline please contact the editorial office immediately.

on behalf of Professor Joshua Ross (Associate Editor) and Mark Chaplain (Subject Editor)
openscience@royalsociety.org

Reviewer comments to Author:

Reviewer: 1

Comments to the Author(s)

I really appreciate the thoroughness of this revision and I wish to thank the authors for their efforts and for taking into accounts my comments: now the paper is definitely much clearer and easier to read and apply!

Just some small things:

- Figure 1 was missing from this version (but present in the track changes one)

- 4:48: Y_{ij}

- 5:28: "matrix whose ... row of Y": this is unclear

- Figure 2: "a wide class of model" -> models

- 5:6 "is a the Euclidean"

- 6:43: "used for as a "

- linguistic example: I think there's a problem with diacritics? I.e. German "green" and Danish "green" seem to miss their vowel (ü and ø respectively)

- 14:4 Mantel test: I am being pedantic, but Mantel's test does seem to have issues

(<https://besjournals.onlinelibrary.wiley.com/doi/10.1111/2041-210x.12018>) -- I don't think that affects your argument here, but still...

===PREPARING YOUR MANUSCRIPT===

one version should clearly identify all the changes that have been made (for instance, in coloured highlight, in bold text, or tracked changes);

===PREPARING YOUR REVISION IN SCHOLARONE===

-- If you are requesting an article processing charge waiver, you must select the relevant waiver option (if requesting a discretionary waiver, the form should have been uploaded, see 'File upload' above).

-- If you have uploaded any electronic supplementary (ESM) files, please ensure you follow the guidance at <https://royalsociety.org/journals/authors/author-guidelines/#supplementary-material> to include a suitable title and informative caption. An example of appropriate titling and captioning may be found at https://figshare.com/articles/Table_S2_from_Is_there_a_trade-off_between_peak_performance_and_performance_breadth_across_temperatures_for_aerobic_scope_in_teleost_fishes_/3843624.

Author's Response to Decision Letter for (RSOS-202182.R1)

See Appendix B.

Decision letter (RSOS-202182.R2)

Dear Dr Lawson,

I am pleased to inform you that your manuscript entitled "CLARITY -- Comparing heterogeneous data using dissimiLARITY" is now accepted for publication in Royal Society Open Science.

on behalf of Professor Joshua Ross (Associate Editor) and Mark Chaplain (Subject Editor)
openscience@royalsociety.org

Appendix A

Dear Authors,

I am in agreement with the Reviewer, that this manuscript makes a valuable contribution but that to ensure its uptake, major revisions are required.

The manuscript is heavy in terminology and concepts, and attempting to make these clearer, **especially earlier in the paper**, would be valued.

We take this feedback seriously and agree that making sure that the reader is with us throughout is vital. We have therefore been much more explicit in the manuscript about both concepts and terminology, and provide summaries at appropriate points. This does overall increase the length (which we now see was too curt for the intended audience). The advantage is that we are now able to do a lot more at the conceptual level with examples, as well as provide "best practice" in a single location, rather than sprinkled throughout the examples. We make the complex terminology clear in "Box 1". We hope that you agree that the ms is significantly easier to approach for a non-specialist, and we are very grateful for the thoughts provided by yourself and the reviewer that led to these major improvements.

Main overarching requests:

1. Break down the examples further, and define explicitly for the examples, what each of the terms/concepts are and correspond to. For example, what are the subjects, what are the features, what are the structures, etc.

This has been done in a thorough revision for each of the sections. We added a sentence to the beginning of the section to introduce the subjects and structures, and explain how the similarity is generated. The features themselves are not so important, as is now made clear.

2. Clearly outline / provide an overview of the two methods -- how they fit in to the overall CLARITY algorithm, and how they differ.

We have added a whole section to address this. It is addressed in the "Structure" section of 2.1, and then described in detail in the new "best practice" section 3.1.

Minor requests:

1. 'similarity' is very loose in 1.1.

We've tried to make this important section easier to understand. The loose use in this section is intentional as we have not yet defined it, but we have tightened the use of similarity and (dis)similarity throughout.

2. It appears that k and K are both used; and k/K are referred to as complexity/components/features. Perhaps attempt to clarify, and be consistent in term usage.

Corrected.

3. Use CLARITY (and not Clarity) throughout.

Corrected.

4. In 2.1, 'subjects' d is introduced, without a precise definition/example.

5. In 2.1, 'structures' A_k is introduced, without a precise definition/example.

6. In 2.1, " X_1 " and " X_2 " appear, without definition. Perhaps the earlier X should be X_1 to better link to Y_1 ?

These are now all defined and provided with examples to aid the reader in this important section, linked to a new visual example in Figure 1 (linked to a reviewer request below). X_1 etc are left for definition later.

7. It feels a bit confusing, in that you state something is appropriate if they are structurally similar; but is that not what you are attempting to assess?

We have tried to make this whole discussion much more thorough and clear throughout; Box 1 contains an explicit definition of Structural Comparison.

Best regards,
Joshua Ross.

Reviewer comments to Author:

Reviewer: 1

Comments to the Author(s)

Overall, I found the paper very interesting and the method potentially very useful, especially for datasets that come from several scientific disciplines and/or are generated using different methodologies. However, I do think the paper needs some work before it can be published in Open Science, especially in what concerns the way it is presented given its potential audience.

We thank you for your care and attention, and have worked hard to act on the spirit of your review, by making the manuscript clearer in structure and more useful as a guide to using our software in practice.

More precisely, I found the paper *very hard to read and digest*, on the one hand, and not uniform in terms of writing (I have the impression that different parts are clearly written by different people having different audiences in mind without a final "harmonization" of the manuscript). This is particularly troubling as it might impact the adoption of the method across disciplines, on the one hand, and will certainly muddle its interpretation, on the other. I hope that this rather general comment might become clearer later on...

We have attempted to address the readability problem with much clearer exposition. The reviewer is very perceptive in that different parts were originally prepared by different people. Revising, we have tried our best to mitigate this. In particular, we re-wrote much of the text in a more uniform style, and several authors made several passes through the text to increase uniformity of exposition. We are happy that we did: it is not for us to judge whether we succeeded in *fully* addressing the present concern, but we ourselves like the current text much better.

Now, to more specific issues:

- number of examples in the paper: the abstract says 3, the introduction says 2; while I know what this means *after* reading the paper, it does not inspire trust

Good point. This has been addressed in the revision, with the example in Figure 1 moved to its own full example, which we believe is important for understanding.

- Figure 1: this is supposed to "ease people" into the paper and summarize what the method is about but I found it very confusing and hard to understand. For example, in panel (b), subpanel (i) it is unclear what the color patches next to the tree are and again if they correspond to those in subpanel (ii). The text says "Data samples are generated as observed with features, belonging to clusters under a 'true structure', all hierarchical clusterings in the data." but this is hard to parse and hard to understand given the amount of information given so far (it becomes clear after reading the paper, but it might be too late by then). Minor things: "For" -> "for" and I guess small k and big K are the same thing?

We have a totally new Figure 1, with a simpler context of clustering. We hope that this is now clear. We have moved the old figure 1 into two parts, figure 2 which covers the simulation and explains what complexity is, and figure 5, the example of methylation.

- p5, l124: "relationship between structures" - this is confusing here and should be much better explained

This section has been rewritten to address yours and the editor's points.

- the linguistic example: while I do have some questions about it (more later), the main gripe I have here is that, as opposed to the cultural one, there is no attempt at analyzing particular languages so as the reader can really see the usefulness of the method besides of "look, differences that you wouldn't see with the naked eye!" Personally, I find it very intriguing that the two directions of prediction look relatively similar but yet so different, and I would have liked some attempted comment on why, for example, the Romance languages (except Romanian) are so deviant and the Germanic ones (except English) when precisely Romanian and English should be relatively "weird" given their history of contact with other languages.

There is a substantive reason why we did it this way: we believe that in this linguistic case study, the scientific significance lies not in the particular languages, but in a global mismatch between phonetic and lexical change historical processes. That said, our revision does discuss specific language groups. Another way we addressed this is that the revised linguistic and culture-economic examples now much more explicitly discuss what methodological and substantive lessons we believe they contribute.

- the cultural example: first, why would one predict culture from economy? I would have naively expected the exact opposite direction.

We believe we worded this example in a misleading way in the submitted version. CLARITY, by construction, is not a tool sensitive to causality as such: it works with statistical dependence; a bit simplistically, one can say, with “correlation”, not with causation. We revised the subsection accordingly. In substance, what we are doing here is identifying countries that have “anomalous” culture conditional on economics. One can also look in the other direction, but we think for the current paper, there is little to gain from examining both: the purpose of this example is to illustrate how to look at the CLARITY residuals for identifying interesting facts in the data.

We agree with the reviewer that there is evidence that cultural drives economic change, especially over the long term. But conversely, economics can result in short term cultural change, and the model is not making a causal claim. Finally, only 9 PCs mean that we cannot apply this approach without reverting to the full CVS dataset. This has all been made explicit in the section.

However, I found this example very well explained and it actually helped me get my head around the outputs of the method! Nevertheless, I was a bit surprised by the interpretation of Andorra (why the Middle East???) but, on the other hand, it was good to see Poland singled out (but I would have naively expected the Czech Republic and Hungary and even East Germany -- if it was there -- to show a similar deviation).

Yes, this is interesting. Czech Republic does indeed have persistent residuals, but the interpretation is more complex, as unlike for Poland, Czech Republic Culturally clusters with Western Europe – and the residuals show interesting structure. There is a paragraph about this now (restricted to where the interpretation is clear).

- discussion: I have to say that I still do not know how to interpret the deviations, which makes it even more frustrating given that this is said to be a major advantage of the method.

We have added an explicit section (Best Practice) on how to interpret residuals (which we confess is not always a trivial problem) and have added interpretation throughout.

Moreover, even if I think I know why, the constant flip in the manuscript between "it's about similarities" and "it's about differences" makes the reader dizzy.

This is still hard to address because we do use both. We hope that the dizziness is reduced by the much more careful introduction to the methods.

- methods: on page 3:45-47 it is said "Unlike distances, dissimilarities need not be symmetric, nor satisfy other useful properties such as the triangle inequality." but then on p13:343 it is said "a dissimilarity matrix is defined to be any symmetric matrix"

We've rewritten the intro to not mention asymmetry, and the Best Practice section now addresses this, and the Methods states "In practice, the utility of the subsequent results does depend on how the matrix was constructed and the 'best practice' input is likely to be a distance matrix corresponding to a reasonable metric, such as Euclidean."

- methods: p14:367-368: the clear statement and description of the *2* methods should come much earlier in the paper -- I was really confused as I thought there were two methods but then I wasn't fully sure

This is now explicit in Section 2.1.

- methods: statistical significance: "However, on the level of subjects, we expect the formally significant results to be scientifically significant" -- I'll be a complete bore, but I am afraid there's quite little overlap between statistical and scientific significance, and what such p-values can do, at best, is signal places where it *might* be worth insisting (on the other hand, for such a method, also places with clearly n.s. p-values might be as equally worth looking at). So, just as a naive suggestion: maybe better to simply show somehow the range/distribution of the resampled values versus the empirical one (as, e.g., sort of stacked series of intervals or such) and let the people decide?

This comment was intended to be about power. We're very sorry to have given this impression, because we totally agree with the reviewer that these are different: "We provide a formal statistical significance procedure to add weight to the identification of scientifically significant results, but emphasise that these are different concepts."

The plots that the reviewer suggests are available but become so very quickly out of hand due to the need to show N^2 histograms that it is hard to recommend them except on very small data examples.

Furthermore, we are now explicitly discussing statistical vs. scientific significance in the two real-world examples (language and culture-economics). We are grateful for this concern, as we believe it spurred us in a very useful presentation direction.

- p18:457: I think something is missing between d and N (maybe "~" ?)

Corrected.

- methods: language example: I am afraid I am not familiar with the particular method developed by Dellert and colleagues, but, naively, it strikes me that maybe one can also use expert cognacy judgments available for the basic vocabulary e.g., in IE, Uralic, Bantu... languages and see how that looks like? It is not that I do not trust the method itself, but I feel relatively unhappy to use it as an *example* for validating a *new method* -- I would have expected to use cases where everybody knows what's going on or, at least, where surprising results cannot be easily relegated to process producing the data

We understand the reviewer's concern and have addressed it in multiple ways. First, the differences between the two types of change are indeed discussed in the linguistic literature, and we included a brief reference to one of the classic sources on that. However, this is, to our knowledge, always made in a qualitative manner, and is based on a small selection of actual linguistic features. Therefore, it is hard to know whether there are really useful generalizations true of the real world. What we do here is novel in that it is a quantitative experiment, including explicit hypothesis testing. As such, there is no existing knowledge that is already available to check our example against. We agree with the reviewer that this means we cannot check if the method is sensible. We believe, though, that our mathematical arguments and our experiments on simulated data reported before the language example are convincing enough in this respect. For us, the value of this example is partly in demonstrating how the new method can be employed to obtain a novel result that was not really possible with any previously designed method.

The second point concerns using automatic cognacy judgements. There are several reasons why we opted for them, some of which we made more prominent in the main text. First, on the level of closely related languages, the cognacy detection is fairly accurate, as one can see from the table (a) in Figure 7 that we added. Second, one "problem" of automatic cognacy detection compared to expert-prepared datasets is that it captures both truly inherited cognates and borrowings, but for us, this is a welcome feature because we want to work with a representation that carries signal both of the underlying backbone tree and of subsequent "horizontal" language contact. Third, for the statistical purposes, it is crucial to have a large set of features, which in this case are meanings. Automatic methods allow us to use the NorthEuraLex dataset with 1016 meanings. Available Indo-European datasets only provide a couple of hundred. The Uralic dataset from Syrjänen et al. contains 226 meanings. The fullest Bantu dataset, by Grollemund et al., has >400 languages, but only 100 meanings (besides, though the cognacy judgements in it are expert-derived, they are not always made based on a full reconstruction; see the comments by Grollemund – we believe she discusses this more in her dissertation than in Grollemund et al. 2015.) So the trade-off is not just between automatic vs. expert-derived: it is also between having ~1000 features vs. having ~200-300 features at most. Even when *individual* features in the expert-derived datasets are more accurate than automatic cognacy judgements, the *whole estimate* of the true similarity matrix would be less stable (technically, would have higher variance) with fewer features. On that point, we effectively check the stability of our estimates of the similarity matrix in the cross-validation procedure. It shows that when the data are divided into two non-overlapping halves, each half very successfully predicts the other, and the variance of our matrix estimates is tolerable. We would not expect this to happen with just 200-300 features.

Thus, what I would suggest is:

1. use a completely made-up *toy example* that is easy to grasp for Figure 1 and also show, graphically, with boxes and arrows, what the method *does* and how to *interpret* its output (kind of what was done for the cultural example, maybe even use this one?)

A very good idea. This has now been done in the form of a simple clustering example.

2. clarify the main text with a non-technical audience in mind

This has been done and we hope that we were successful!

3. for the linguistic example, maybe use a more "standard" cognacy judgment (maybe on a reduced set?) and *explain* what the output *means* given what we know about the history of those languages

Addressed, though in a different way than suggested here – see our remarks above.

Finally we would like to thank you for your careful attention and valuable feedback!

Appendix B

We have been able to complete all of the actions required and would like to thank both the reviewer, and the associate editor, for their very valuable contributions to this work.

Comments to the Author(s)

I really appreciate the thoroughness of this revision and I wish to thank the authors for their efforts and for taking into account my comments: now the paper is definitely much clearer and easier to read and apply!

Thank you, we really appreciate this very careful and helpful review!

Just some small things:

- Figure 1 was missing from this version (but present in the track changes one)

Presumably an issue with the manuscript central compilation, we will ensure that it is present in this updated version.

- 4:48: Y_{ij}

Fixed.

- 5:28: "matrix whose ... row of Y": this is unclear

This sentence was restructured to make this clear.

- Figure 2: "a wide class of model" -> models

Fixed.

- 5:6 "is a the Euclidean"

Fixed.

- 6:43: "used for as a "

Fixed.

- linguistic example: I think there's a problem with diacritics? I.e. German "green" and Danish "green" seem to miss their vowel (ü and ø respectively)

This appears to be a Unicode problem, as they appeared correctly for us. We have removed the Unicode and replaced with latex codes, which should address the issue.

- 14:4 Mantel test: I am being pedantic, but Mantel's test does seem to have issues (<https://besjournals.onlinelibrary.wiley.com/doi/10.1111/2041-210x.12018>) -- I don't think that affects your argument here, but still...

We agree! The reference makes the point clearly. Without wanting to get sidetracked, we have simply removed it from our example, though it remains discussed in 1.2 in a way that is consistent with the reference.